# RETHINKING PSEUDO-LABELING: DATA-CENTRIC INSIGHTS IMPROVE SEMI-SUPERVISED LEARNING

## ABSTRACT

Pseudo-labeling is a popular semi-supervised learning technique to leverage unlabeled data when labeled samples are scarce. The generation and selection of pseudo-labels heavily rely on labeled data. Existing approaches implicitly assume that the labeled data is gold standard and "perfect". However, this can be violated in reality with issues such as mislabeling or ambiguity. We address this overlooked aspect and show the importance of investigating *labeled data* quality to improve *any* pseudo-labeling method. Specifically, we introduce a novel data characterization and selection framework called `DIPS` to extend pseudo-labeling. We select useful labeled and pseudo-labeled samples via analysis of learning dynamics. We demonstrate the applicability and impact of `DIPS` for various pseudo-labeling methods across an extensive range of real-world tabular and image datasets. Additionally, `DIPS` improves data efficiency and reduces the performance distinctions between different pseudo-labelers. Overall, we highlight the significant benefits of a data-centric rethinking of pseudo-labeling in real-world settings.

## 1 INTRODUCTION

Machine learning heavily relies on the availability of large numbers of annotated training examples. However, in many real-world settings, such as healthcare and finance, collecting even limited numbers of annotations is often either expensive or practically impossible. Semi-supervised learning leverages unlabeled data to combat the scarcity of labeled data (Zhu, 2005; Chapelle et al., 2006; van Engelen & Hoos, 2019). Pseudo-labeling is a prominent semi-supervised approach applicable across data modalities that assigns pseudo-labels to unlabeled data using a model trained on the labeled dataset. The pseudo-labeled data is then combined with labeled data to produce an augmented training set. This increases the size of the training set and has been shown to improve the resulting model. In contrast, consistency regularization methods (Sohn et al., 2020) are less versatile and often not applicable to settings such as tabular data, where defining the necessary semantic-preserving augmentations proves challenging (Gidaris et al., 2018; Nguyen et al., 2022a). Given the broad applicability across data modalities and competitive performance, we focus on pseudo-labeling approaches.

**Labeled data is not always gold standard.** Current pseudo-labeling methods focus on unlabeled data selection. However, an equally important yet *overlooked* problem is around labeled data quality, given the reliance of pseudo-labelers on the labeled data. In particular, it is often implicitly assumed that the labeled data is "gold standard and perfect". This "gold standard" assumption is unlikely to hold in reality, where data can have issues such as mislabeling and ambiguity (Sambasivan et al., 2021; Renggli et al., 2021; Jain et al., 2020; Gupta et al., 2021a;b; Northcutt et al., 2021a;b). For example, Northcutt et al. (2021b) quantified the label error rate of widely-used benchmark datasets, reaching up to 10%, while Wei et al. (2022a) showed this can be as significant as 20-40%. This issue is critical for pseudo-labeling, as labeled data provides the supervision signal for pseudo-labels. Hence, issues in the labeled data will affect the pseudo-labels and the predictive model (see Fig. 1). Mechanisms to address this issue are essential to improve pseudo-labeling. It might appear possible to manually inspect the data to identify errors in the labeled set. However, this requires domain expertise and is human-intensive, especially in modalities such as tabular data where inspecting rows in a spreadsheet can be much more challenging than reviewing an image. In other cases, updating labels might be infeasible due to rerunning costly experiments in domains such as biology and physics, or indeed impossible due to lack of access to either the underlying sample or equipment.

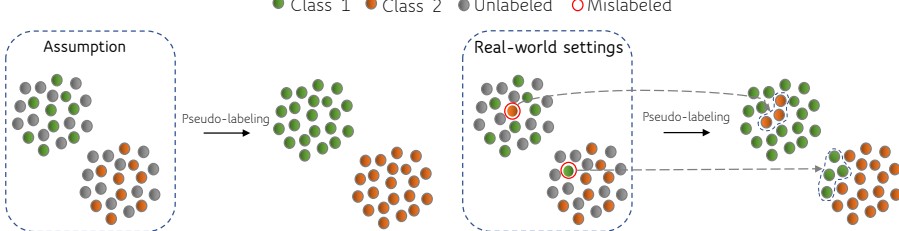

Figure 1: **(Left)** Current pseudo-labeling formulations implicitly assume that the labeled data is the gold standard. **(Right)** However, this assumption is violated in real-world settings. Mislabeled samples lead to error propagation when pseudo-labeling the unlabeled data.

**Extending the pseudo-labeling machinery.** To solve this fundamental challenge, we propose a novel framework to extend the pseudo-labeling machinery called **D**ata-centric **I**nsights for **P**seudo-labeling with **S**election (**DIPS**). `DIPS` focuses on the labeled and pseudo-labeled data to characterize and select the most useful samples. We instantiate `DIPS` based on learning dynamics — the behavior of individual samples during training. We analyze the dynamics by computing two metrics, confidence and aleatoric (data) uncertainty, which enables the characterization of samples as *Useful* or *Harmful*, guiding sample selection for model training. Sec. 5 empirically shows that this selection improves pseudo-labeling performance in multiple real-world settings.

Beyond performance, `DIPS` is also specifically designed to be a flexible solution that easily integrates with existing pseudo-labeling approaches, having the following desired properties:

> **(P1) Plug & Play:** applicable on top of *any* pseudo-labeling method (to improve it).
> **(P2) Model-agnostic data characterization:** agnostic to any class of supervised backbone models trained in an iterative scheme (e.g. neural networks, boosting methods).
> **(P3) Computationally cheap:** minimal computational overhead to be practically usable.

**Contributions:** ① *Conceptually*, we propose a rethinking of pseudo-labeling, demonstrating the importance of characterizing and systematically selecting data from both the labeled and pseudo-labeled datasets, in contrast to the current focus *only* on the unlabeled data. ② *Technically*, we introduce `DIPS`, a novel framework to characterize and select the most useful samples for pseudo-labeling. This extends the pseudo-labeling machinery to address the unrealistic status quo of considering the labeled data as gold standard. ③ *Empirically*, we show the value of taking into account labeled data quality, with DIPS's selection mechanism improving various pseudo-labeling baselines, both in terms of *performance* and *data efficiency*, which we demonstrate across 18 real-world datasets, spanning both tabular data and images. This highlights the usefulness and applicability of `DIPS`.

## 2  RELATED WORK

**Semi-supervised learning and pseudo-labeling methods.** Semi-supervised learning leverages unlabeled data to combat the scarcity of labeled data (Zhu, 2005; Chapelle et al., 2006; van Engelen & Hoos, 2019; Iscen et al., 2019; Berthelot et al., 2019). As mentioned in Sec. 1, we focus on pseudo-labeling approaches, given their applicability across data modalities and competitive performance. Recent methods have extended pseudo-labeling by modifying the selection mechanism of unlabeled data (Lee et al., 2013; Rizve et al., 2021; Nguyen et al., 2022a; Tai et al., 2021), using curriculum learning (Cascante-Bonilla et al., 2020), or merging pseudo-labeling with consistency loss-focused regularization (Sohn et al., 2020). A commonality among these works is a focus on ensuring the correct selection of the unlabeled data, assuming a gold standard labeled data. In contrast, `DIPS` addresses the question: "What if the labeled data is not gold standard?", extending the aforementioned approaches to be more performant.

**Data-centric AI.** Data-centric AI has emerged focused on developing systematic methods to improve the quality of data (Liang et al., 2022). One aspect is to score data samples based on their utility for a task, or whether samples are easy or hard to learn, then enabling the curation or sculpting of high-quality datasets for training efficiency purposes (Paul et al., 2021) or improved performance (Liang et al., 2022). Typically, the goal is to identify mislabeled, hard, or ambiguous examples, with methods differing based on metrics including uncertainty (Swayamdipta et al., 2020; Seedat et al., 2022), logits

(Pleiss et al., 2020), gradient norm (Paul et al., 2021), or variance of gradients (Agarwal et al., 2022). We note two key aspects: (1) we draw inspiration from their success in the fully supervised setting (where there are large amounts of labeled data) and bring the idea to the semi-supervised setting where we have unlabeled data but scarce labeled data; (2) many of the discussed supervised methods are only applicable to neural networks, relying on gradients or logits. Hence, they are not broadly applicable to any model class, such as boosting methods which are predominant in tabular settings (Borisov et al., 2021; Grinsztajn et al., 2022). This violates **P2: Model-agnostic data characterization**.

**Learning with Noisy Labels (LNL).** LNL typically operates in the supervised setting and assumes access to a large amount of labeled data. This contrasts the semi-supervised setting, where labeled data is scarce, and is used to output pseudo-labels for unlabeled data. Some LNL methods alter a loss function, e.g. adding a regularization term (Cheng et al., 2021; Wei et al., 2022b). Other methods select samples using a uni-dimensional metric, the most common being the small-loss criterion in the supervised setting (Xia et al., 2021). `DIPS` contrasts these approaches by taking into account both confidence and aleatoric uncertainty in its selection process. While the LNL methods have not been used in the semi-supervised setting previously, we repurpose them for this setting and experimentally highlight the value of the curation process of `DIPS` in Appendix C. Interestingly, pseudo-labeling can also be used as a tool in the supervised setting to relabel points identified as noisy by treating them as unlabeled (Li et al., 2019); however, this contrasts DIPS in two key ways: (1) data availability: these works operate *only* on large labeled datasets, whereas DIPS operates with a small labeled and large unlabeled dataset. (2) application: these works use pseudo-labeling as a tool for supervised learning, whereas DIPS extends the machinery of pseudo-labeling itself.

## 3 BACKGROUND

We now give a brief overview of pseudo-labeling as a general paradigm of semi-supervised learning. We then highlight that the current formulation of pseudo-labeling overlooks the key notion of labeled data quality, which motivates our approach.

### 3.1 SEMI-SUPERVISED LEARNING VIA PSEUDO-LABELING

Semi-supervised learning addresses the scarcity of labeled data by leveraging unlabeled data. The natural question it answers is: how can we combine labeled and unlabeled data in order to boost the performance of a model, in comparison to training on the small labeled data alone?

**Notation.** Consider a classification setting where we have a labeled dataset $\mathcal{D}_{\text{lab}} = \{(x_i, y_i) | i \in [n_{\text{lab}}]\}$ as well as an unlabeled dataset $\mathcal{D}_{\text{unlab}} = \{x'_j | j \in [n_{\text{unlab}}]\}$. We typically assume that $n_{\text{lab}} \ll n_{\text{unlab}}$. Moreover, the labels take values in $\{0, 1\}^C$, where $C$ is the number of classes. This encompasses both binary ($C = 2$) and multi-label classification. Our goal is to learn a predictive model $f : x \to y$ which leverages $\mathcal{D}_{\text{unlab}}$ in addition to $\mathcal{D}_{\text{lab}}$, such that it performs better than a model trained on the small labeled dataset $\mathcal{D}_{\text{lab}}$ alone. For all $k \in [C]$, the $k$-th coordinate of $f(x)$ is denoted as $[f(x)]_k$. It is assumed to be in $[0, 1]$, which is typically the case after a softmax layer.

**Pseudo-labeling.** Pseudo-labeling (PL) is a powerful and general-purpose semi-supervised approach which answers the pressing question of how to incorporate $\mathcal{D}_{\text{unlab}}$ in the learning procedure. PL is an iterative procedure which spans $T$ iterations and constructs a succession of models $f^{(i)}$, for $i = 1, ..., T$. The result of this procedure is the last model $f^{(T)}$, which issues predictions at test time. The idea underpinning PL is to gradually incorporate $\mathcal{D}_{\text{unlab}}$ into $\mathcal{D}_{\text{lab}}$ to train the classifiers $f^{(i)}$. At each iteration $i$ of pseudo-labeling, two steps are conducted in turn. *Step 1:* The model $f^{(i)}$ is first trained with supervised learning. *Step 2:* $f^{(i)}$ then pseudo-labels unlabeled samples, a subset of which are selected to expand the training set of the next classifier $f^{(i+1)}$. The key to PL is the construction of these training sets. More precisely, let us denote $\mathcal{D}_{\text{train}}^{(i)}$ the training set used to train $f^{(i)}$ at iteration $i$. $\mathcal{D}_{\text{train}}^{(i)}$ is defined by an initial condition, $\mathcal{D}_{\text{train}}^{(1)} = \mathcal{D}_{\text{lab}}$, and by the following recursive equation: for all $i = 1, ..., T - 1, \mathcal{D}_{\text{train}}^{(i+1)} = \mathcal{D}_{\text{train}}^{(i)} \cup s(\mathcal{D}_{\text{unlab}}, f^{(i)})$, where $s$ is a selector function. Alternatively stated, $f^{(i)}$ outputs pseudo-labels for $\mathcal{D}_{\text{unlab}}$ at iteration $i$ and the selector function $s$ then selects a subset of these pseudo-labeled samples, which are added to $\mathcal{D}_{\text{train}}^{(i)}$ to form $\mathcal{D}_{\text{train}}^{(i+1)}$. Common heuristics define $s$ with metrics of confidence and/or uncertainty (e.g. greedy-PL (Lee et al., 2013), UPS (Rizve et al., 2021)). More details are given in Appendix A regarding the exact formulation of $s$ in those cases.

### 3.2 Overlooked aspects in the current formulation of pseudo-labeling

Having introduced the pseudo-labeling paradigm, we now show that its current formulation overlooks several key elements which will motivate our approach.

First, the selection mechanism $s$ only focuses on unlabeled data and ignores labeled data. This implies that the labeled data is considered "perfect". This assumption is not reasonable in many real-world settings where labeled data is noisy. In such situations, as shown in Fig. 1, noise propagates to the pseudo-labels, jeopardizing the accuracy of the pseudo-labeling steps (Nguyen et al., 2022a). To see why such propagation of error happens, recall that $\mathcal{D}_{\text{train}}^{(1)} = \mathcal{D}_{\text{lab}}$. Alternatively stated, $\mathcal{D}_{\text{lab}}$ provides the initial supervision signal for PL and its recursive construction of $\mathcal{D}_{\text{train}}^{(i)}$.

Second, PL methods do not update the pseudo-labels of unlabeled samples once they are incorporated in one of the $\mathcal{D}_{\text{train}}^{(i)}$. However, the intuition underpinning PL is that the classifiers $f^{(i)}$ progressively get better over the iterations, meaning that pseudo-labels computed at iteration $T$ are expected to be more accurate than pseudo-labels computed at iteration 1, since $f^{(T)}$ is the output of PL.

Taken together, these two observations shed light on an important yet overlooked aspect of current PL methods: the selection mechanism $s$ ignores labeled and previously pseudo-labeled samples. This naturally manifests in the asymmetry of the update rule $\mathcal{D}_{\text{train}}^{(i+1)} = \mathcal{D}_{\text{train}}^{(i)} \cup s(\mathcal{D}_{\text{unlab}}, f^{(i)})$, where the selection function $s$ is only applied to *unlabeled data* and ignores $\mathcal{D}_{\text{train}}^{(i)}$ at iteration $i + 1$.

## 4 DIPS: Data-centric insights for improved pseudo-labeling

In response to these overlooked aspects, we propose a new formulation of pseudo-labeling, DIPS, with the data-centric aim to characterize the usefulness of both *labeled* and *pseudo-labeled* samples. We then operationalize this framework with the lens of learning dynamics. Our goal is to improve the performance of *any* pseudo-labeling algorithm by selecting *useful* samples to be used for training.

### 4.1 A data-centric formulation of pseudo-labeling

Motivated by the asymmetry in the update rule of $\mathcal{D}_{\text{train}}^{(i)}$, as defined in Sec. 3.1, we propose DIPS, a novel framework which explicitly focuses on both labeled and pseudo-labeled samples. The key idea is to introduce a new selection mechanism, called $r$, while still retaining the benefits of $s$. For any dataset $\mathcal{D}$ and classifier $f$, $r(\mathcal{D}, f)$ defines a subset of $\mathcal{D}$ to be used for training in the current pseudo-labeling iteration. More formally, we define the new update rule (Eq. 1) for all $i = 1, ..., T - 1$ as:

$$\begin{cases} \mathcal{D}^{(i+1)} &= \mathcal{D}^{(i)} \cup s(\mathcal{D}_{\text{unlab}}, f^{(i)}) \quad \triangleright \text{Original PL formulation} \\ \mathcal{D}_{\text{train}}^{(i+1)} &= r(\mathcal{D}^{(i+1)}, f^{(i)}) \quad\quad\quad \triangleright \text{DIPS selection} \end{cases} \quad (1)$$

Then, let $\mathcal{D}^{(1)} = \mathcal{D}_{\text{train}}^{(1)} = r(\mathcal{D}_{\text{lab}}, f^{(0)})$, where $f^{(0)}$ is a classifier trained on $\mathcal{D}_{\text{lab}}$ only. The selector $r$ selects samples from $\mathcal{D}^{(i+1)}$, producing $\mathcal{D}_{\text{train}}^{(i+1)}$, the training set of $f^{(i+1)}$.

This new formulation addresses the challenges mentioned in Sec. 3.2, as it considers both *labeled* and *pseudo-labeled* samples. Indeed, $r$ selects samples in $\mathcal{D}_{\text{lab}}$ at any iteration $i$, since $r(\mathcal{D}_{\text{lab}}, f^{(0)}) = \mathcal{D}^{(0)} \subset \mathcal{D}^{(i)}$ for all $i = 0, ..., T$. Moreover, at any iteration $j = 2, ..., T$, $r$ selects samples amongst those pseudo-labeled at a previous iteration $i < j$, since $s(\mathcal{D}_{\text{unlab}}, f^{(i-1)}) \subset \mathcal{D}^{(i)} \subset \mathcal{D}^{(j)}$. We show the value of considering both labeled and pseudo-labeled data for the selection in Appendix C.1.

Finally, notice that DIPS subsumes current pseudo-labeling methods via its selector $r$. To see that, we note current pseudo-labeling methods define an identity selector $r$, selecting all samples, such that for any $\mathcal{D}$ and function $f$, we have $r(\mathcal{D}, f) = \mathcal{D}$. Hence, DIPS goes beyond this status quo by permitting a non-identity selector $r$.

### 4.2 Operationalizing DIPS using learning dynamics

We now explicitly instantiate DIPS by constructing the selector $r$. Our key idea is to define $r$ using learning dynamics of samples. Before giving a precise definition, let us detail some context. Prior works in *learning theory* have shown that the learning dynamics of samples contain a useful signal about the nature of samples (and their usefulness) for a specific task (Arpit et al., 2017; Arora et al.,

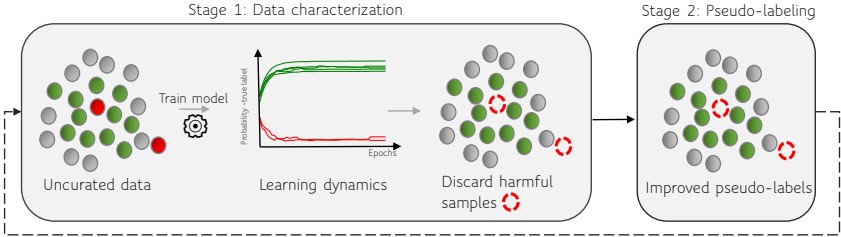

Figure 2: Stage 1 operationalizes DIPS by leveraging learning dynamics of individual labeled and pseudo-labeled samples to characterize them as *Useful* or *Harmful*. Only Useful samples are then kept for Stage 2, which consists of pseudo-labeling, using any off-the-shelf method.

2019; Li et al., 2020). Some samples may be easier for a model to learn, whilst for other samples, a model might take longer to learn (i.e. more variability through training) or these samples might not be learned correctly during the training process. We build on this insight about learning dynamics, bringing the idea to the *semi-supervised setting*.

Our construction of $r$ has three steps. First, we analyze the learning dynamics of labeled and pseudo-labeled samples to define two metrics: (i) confidence and (ii) aleatoric uncertainty, which captures the inherent data uncertainty. Second, we use these metrics to characterize the samples as *Useful* or *Harmful*. Third, we select *Useful* samples for model training, which gives our definition of $r$.

For any $i$, we assume that the classifier $f^{(i)}$ at iteration $i$ of PL is trained in an iterative scheme (e.g. neural networks or XGBoost trained over iterations), which is ubiquitous in practice. This motivates and makes it possible to analyze the learning dynamics as a way to characterize individual samples. For clarity of presentation, we consider binary classification ($C = 2$) and denote $f^{(i)} = f$.

At any pseudo-labeling iteration, $f$ is trained from scratch and goes through $e \in [E]$ different checkpoints leading to the set $\mathcal{F} = \{f_1, f_2, \ldots, f_E\}$, such that $f_e$ is the classifier at the $e$-th checkpoint. Our goal is to assess the learning dynamics of samples over these $E$ training checkpoints. For this, we define $H$, a random variable following a uniform distribution $\mathcal{U}_\mathcal{F}$ over the set of checkpoints $\mathcal{F}$. Specifically, given $H = h$ and a sample $(x, y)$, where $y$ is either a provided label ($x \in \mathcal{D}_{\text{lab}}$) or a pseudo-label ($x \in \mathcal{D}_{\text{unlab}}$), we define the correctness in the prediction of $H$ as a binary random variable $\hat{Y}_\mathcal{F}(x, y)$ with the following conditional: $P(\hat{Y}_\mathcal{F}(x, y) = 1 | H = h) = [h(x)]_y$ and $P(\hat{Y}_\mathcal{F}(x, y) = 0 | H = h) = 1 - P(\hat{Y}_\mathcal{F}(x, y) = 1 | H = h)$.

Equipped with a probabilistic interpretation of the predictions of a model, we now define our characterization metrics: (i) average confidence and (ii) aleatoric (data) uncertainty, inspired by (Kwon et al., 2020; Seedat et al., 2022).

**Definition 4.1** (Average confidence). For any set of checkpoints $\mathcal{F} = \{f_1, ..., f_E\}$, the average confidence for a sample $(x, y)$ is defined as the following marginal:

$$\bar{\mathcal{P}}_\mathcal{F}(x, y) := P(\hat{Y}_\mathcal{F}(x, y) = 1) = \mathbb{E}_{H \sim \mathcal{U}_\mathcal{F}}[P(\hat{Y}_\mathcal{F}(x, y) = 1 | H)] = \frac{1}{E} \sum_{e=1}^{E} [f_e(x)]_y \qquad (2)$$

**Definition 4.2** (Aleatoric uncertainty). For any set of checkpoints $\mathcal{F} = \{f_1, ..., f_E\}$, the aleatoric uncertainty for a sample $(x, y)$ is defined as:

$$v_{al, \mathcal{F}}(x, y) := \mathbb{E}_{H \sim \mathcal{U}_\mathcal{F}}[Var(\hat{Y}_\mathcal{F}(x, y) | H)] = \frac{1}{E} \sum_{e=1}^{E} [f_e(x)]_y (1 - [f_e(x)]_y) \qquad (3)$$

Intuitively, the aleatoric uncertainty for a sample $x$ is maximized when $[f_e(x)]_y = \frac{1}{2}$ for all checkpoints $f_e$, akin to random guessing. Recall aleatoric uncertainty captures the inherent data uncertainty, hence is a principled way to capture issues such as mislabeling. This contrasts epistemic uncertainty, which is model-dependent and can be reduced simply by increasing model parameterization (Hüllermeier & Waegeman, 2021).

We emphasize that this definition of uncertainty is model-agnostic, satisfying **P2: Model-agnostic data characterization**, and only relies on having checkpoints through training. Hence, it comes for *free*, unlike ensembles (Lakshminarayanan et al., 2017). This fulfills **P3: Computationally cheap**. Moreover, it is applicable to any iteratively trained model (e.g. neural networks and XGBoost) unlike approaches such as MC-dropout or alternative training dynamic metrics using gradients (Paul et al., 2021) or logits (Pleiss et al., 2020).

### 4.3 DEFINING THE SELECTOR $r$: DATA CHARACTERIZATION AND SELECTION

Having defined sample-wise confidence and aleatoric uncertainty, we characterize both labeled and pseudo-labeled samples into two categories, namely *Useful* and *Harmful*. Given a sample $(x, y)$, a set of training checkpoints $\mathcal{F}$, and two thresholds $\tau_{\text{conf}}$ and $\tau_{\text{al}}$, we define the category $c(x, y, \mathcal{F})$ as *Useful* if $\bar{\mathcal{P}}_{\mathcal{F}}(x, y) \geq \tau_{\text{conf}}$ and $v_{al, \mathcal{F}}(x, y) < \tau_{\text{al}}$, and *Harmful* otherwise.

Hence, a *Useful* sample is one where we are highly confident in predicting its associated label and for which we also have low inherent data uncertainty. In contrast, a harmful sample would have low confidence and/or high data uncertainty. Finally, given a function $f$ whose training led to the set of checkpoints $\mathcal{F}$, we can define $r$ explicitly by $r(\mathcal{D}, f) = \{(x, y) | (x, y) \in \mathcal{D}, c(x, y, \mathcal{F}) = Useful\}$.

### 4.4 COMBINING DIPS WITH *any* PSEUDO-LABELING ALGORITHM

We outline the integration of DIPS into *any* pseudo-labeling algorithm as per Algorithm 1 (see Appendix A). A fundamental strength of DIPS lies in its simplicity. Not only is computation almost for free (no extra model training) – i.e. **P3: Computationally cheap**, but also DIPS is easily integrated into *any* pseudo-labeling algorithm – i.e. **P1: Plug & Play**, making for easier adoption.

---

**Algorithm 1** Plug DIPS into *any* pseudo-labeler

1: Train a network, $f^{(0)}$, using the samples from $\mathcal{D}_{\text{lab}}$.
2: Plug-in DIPS: set $\mathcal{D}^{(1)}_{\text{train}} = \mathcal{D}^{(1)} = r(\mathcal{D}_{\text{lab}}, f^{(0)})$
3: **for** $t = 1..\text{T}$ **do**
4:     Initialize new network $f^{(t)}$
5:     Train $f^{(t)}$ using $\mathcal{D}^{(t)}_{\text{train}}$.
6:     Pseudo-label $\mathcal{D}_{\text{unlab}}$ using $f^{(t)}$
7:     Define $\mathcal{D}^{(t+1)}$ using the PL method's selector $s$
8:     Plug-in DIPS : Define $\mathcal{D}^{(t+1)}_{\text{train}} = r(\mathcal{D}^{(t+1)}, f^{(t)})$   ▷ Data characterization and selection, Sec. 4.3
9: **end for**
10: **return** $f_T$

---

## 5 EXPERIMENTS

We now empirically investigate multiple aspects of DIPS. We discuss the setup of each experiment at the start of each sub-section, with further experimental details in Appendix B.

1. **Characterization: Does it matter?** Sec. 5.1 analyzes the effect of not characterizing and selecting samples in all $\mathcal{D}^{(i)}_{\text{train}}$ in a synthetic setup, where noise propagates from $\mathcal{D}_{\text{lab}}$ to $\mathcal{D}_{\text{unlab}}$.
2. **Performance: Does it work?** Sec. 5.2 shows characterizing $\mathcal{D}^{(i)}_{\text{train}}$ using DIPS improves performance of various state-of-the-art pseudo-labeling baselines across 12 real-world datasets.
3. **Narrowing the gap: Can selection reduce performance disparities?** Sec. 5.2 shows that DIPS also renders the PL methods more comparable to one other.
4. **Data efficiency: Can similar performance be achieved with less labeled data?** Sec. 5.3 studies the efficiency of data usage of vanilla methods vs. DIPS on different proportions of labeled data.
5. **Selection across countries: Can selection improve performance when using data from a different country?** Sec. 5.4 assesses the role of selection of labeled and pseudo-labeled samples when $\mathcal{D}_{\text{lab}}$ and $\mathcal{D}_{\text{unlab}}$ come from different countries in a clinically relevant task.
6. **Other modalities**: Sec. 5.5 shows the potential to use DIPS in image experiments.

### 5.1 SYNTHETIC EXAMPLE: DATA CHARACTERIZATION AND UNLABELED DATA IMPROVE TEST ACCURACY

**Goal.** To motivate DIPS, we demonstrate that: (1) label noise in the data harms pseudo-labeling (2) characterizing and selecting data using DIPS consequently improves pseudo-labeling performance.

**Setup.** We consider a synthetic setup with two quadrants (Lee et al., 2023), as illustrated in Fig. 3b [1]. We sample data in each of the two quadrants from a uniform distribution, and each sample is equally likely to fall in each quadrant. To mimic a real-world scenario of label noise in $\mathcal{D}_{\text{lab}}$, we randomly flip labels with varying proportions $p_{\text{corrupt}} \in [0.1, 0.45]$

**Baselines.** We compare DIPS with two baselines **(i) Supervised** which trains a classifier using the initial $\mathcal{D}_{\text{lab}}$ **(ii) Greedy pseudo-labeling (PL)** (Lee et al., 2013) which uses both $\mathcal{D}_{\text{lab}}$ and $\mathcal{D}_{\text{unlab}}$. We use an XGBoost backbone for all the methods and we combine DIPS with PL for a fair comparison.

---

[1]Notice that the two quadrant setup satisfies the cluster assumption inherent to the success of semi-supervised learning (Chapelle et al., 2006).

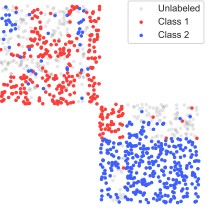 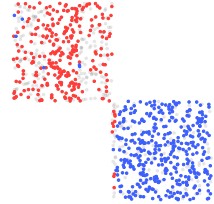 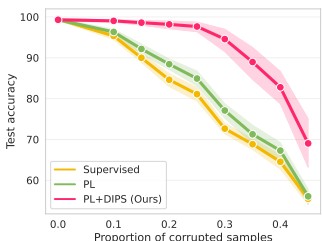

(a) PL: The inherent noise in $\mathcal{D}_{\text{lab}}$ propagates when assigning pseudo-labels.

(b) PL+DIPS: DIPS mitigates the issue of noise by selecting useful samples.

(c) DIPS selection mechanism significantly improves test performance under label noise.

Figure 3: **(Left)** The colored dots are an illustration of the selected labeled and pseudo-labeled samples for the last iteration of PL and PL+DIPS, with $30\%$ label noise. Grey dots are unselected unlabeled samples. **(Right)** Characterizing and selecting data for the semi-supervised algorithm yields the best results, as epitomized by PL+DIPS, and makes the use of unlabeled data impactful.

**Results.** Test performance over 20 random seeds with different data splits and $n_{\text{lab}} = 100, n_{\text{unlab}} = 900$ is illustrated in Fig. 3c, for varying $p_{\text{corrupt}}$. It highlights two key elements. First, PL barely improves upon the supervised baseline. The noise in the labeled dataset propagates to the unlabeled dataset, via the pseudo-labels, as shown in Fig. 3a. This consequently negates the benefit of using the unlabeled data to learn a better classifier, which is the original motivation of semi-supervised learning. Second, DIPS mitigates this issue via its selection mechanism and improves performance by around **+20%** over the two baselines when the amount of label noise is around $30\%$. We also conduct an ablation study in Appendix C to understand when in the pseudo-labeling pipeline to apply DIPS.

**Takeaway.** The results emphasize the key motivation of DIPS: labeled data quality is central to the performance of the pseudo-labeling algorithms because labeled data drives the learning process necessary to perform pseudo-labeling. Hence, careful consideration of $\mathcal{D}_{\text{lab}}$ is crucial to performance.

## 5.2 DIPS IMPROVES DIFFERENT PSEUDO-LABELING ALGORITHMS ACROSS 12 REAL-WORLD TABULAR DATASETS.

**Goal.** We evaluate the effectiveness of DIPS on 12 different real-world tabular datasets with diverse characteristics (sample sizes, number of features, task difficulty). We aim to demonstrate that DIPS improves the performance of various pseudo-labeling algorithms. We focus on the tabular setting, as pseudo-labeling plays a crucial role in addressing data scarcity issues in healthcare and finance, discussed in Sec. 1, where data is predominantly tabular (Borisov et al., 2021; Shwartz-Ziv & Armon, 2022). Moreover, enhancing our capacity to improve models for tabular data holds immense significance, given its ubiquity in real-world applications. For perspective, nearly 79% of data scientists work with tabular data on a daily basis, compared to only 14% who work with modalities such as images (Kaggle, 2017). This underlines the critical need to advance pseudo-labeling techniques in the context of impactful real-world tabular data.

**Datasets.** The tabular datasets are drawn from a variety of domains (e.g. healthcare, finance), mirroring Sec. 1, where the issue of limited annotated examples is highly prevalent. It is important to note that the vast majority of the datasets (10/12) are real-world data sets, demonstrating the applicability of DIPS and its findings in practical scenarios. For example, Covid-19 (Baqui et al., 2020), MAGGIC (Pocock et al., 2013), SEER (Duggan et al., 2016), and CUTRACT (PCUK, 2019) are medical datasets. COMPAS (Angwin et al., 2016) is a recidivism dataset. Credit is a financial default dataset from a Taiwan bank (Yeh & Lien, 2009). Higgs is a physics dataset (Baldi et al., 2014). The datasets vary significantly in both sample size (from 1k to 41k) and number of features (from 12 to 280). More details on the datasets can be found in Table 1, Appendix B.

**Baselines.** As baselines, we compare: (i) **Supervised** training on the small $\mathcal{D}_{\text{lab}}$, (ii) *five* state-of-the-art pseudo-labeling methods applicable to tabular data: **greedy-PL** (Lee et al., 2013), **UPS** (Rizve et al., 2021), **FlexMatch** (Zhang et al., 2021), **SLA** (Tai et al., 2021), **CSA** (Nguyen et al., 2022a). For each of the baselines, we apply DIPS as a plug-in to improve performance.

**Results** We report results in Fig. 4 across 50 random seeds with different data splits with a fixed proportion of $\mathcal{D}_{\text{lab}} : \mathcal{D}_{\text{unlab}}$ of 0.1:0.9. We note several findings from Fig. 4 pertinent to DIPS.

■ **DIPS improves the performance of almost all baselines across various real-world datasets.**
We showcase the value of data characterization and selection to improve SSL performance. We

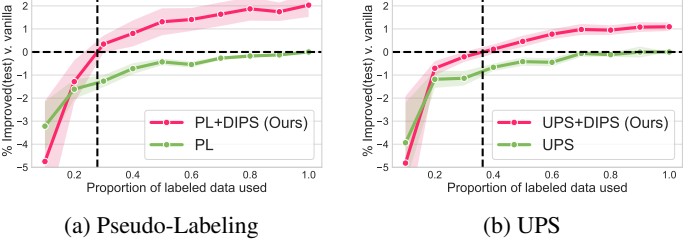

Figure 4: `DIPS` consistently improves performance of all five pseudo-labeling methods across the 12 real-world datasets. `DIPS` also reduces the performance gap between the different pseudo-labelers.

demonstrate that `DIPS` consistently boosts the performance when incorporated with existing pseudo-labelers. This illustrates the key motivation of our work: labeled data is of critical importance for pseudo-labeling and calls for curation, in real-world scenarios.

■ **`DIPS` reduces the performance gap between pseudo-labelers.**
Fig. 4 shows the reduction in variability of performance across pseudo-labelers by introducing data characterization. On average, we reduce the average variance across all datasets and algorithms from 0.46 in the vanilla case to 0.14 using `DIPS`. In particular, we show that the simplest method, namely greedy pseudo-labeling (Lee et al., 2013), which is often the worst in the vanilla setups, is drastically improved simply by incorporating `DIPS`, making it competitive with the more sophisticated alternative baselines. This result of equalizing performance is important as it directly influences the process of selecting a pseudo-labeling algorithm. We report additional results in Appendix C.2 where we replace the selector $r$ with sample selectors from the LNL literature, highlighting the advantage of using learning dynamics.

**Takeaway.** We have empirically demonstrated improved performance by `DIPS` across multiple pseudo-labeling algorithms and multiple real-world datasets.

## 5.3 `DIPS` IMPROVES DATA EFFICIENCY

**Goal.** In real-world scenarios, collecting labeled data is a significant bottleneck, hence it is traditionally the sole focus of semi-supervised benchmarks. The goal of this experiment is to demonstrate that data quality is an overlooked dimension that has a direct impact on data quantity requirements to achieve a given test performance for pseudo-labeling.

**Setup.** For clarity, we focus on greedy-PL and UPS as pseudo-labeling algorithms. To assess data efficiency, we consider subsets of $\mathcal{D}_{\text{lab}}$ with size $p \cdot |\mathcal{D}_{\text{lab}}|$, with $p$ going from 0.1 to 1.

(a) Pseudo-Labeling  (b) UPS

Figure 5: `DIPS` improves data efficiency of vanilla methods, achieving the same level of performance with *60-70%* fewer labeled examples, as shown by the vertical dotted lines. The results are averaged across datasets and show gains in accuracy vs. the maximum performance of the vanilla method.

**Results.** The results in Fig. 5, averaged across datasets, show the performance gain in accuracy for all $p$ compared to the maximum performance of the vanilla method (i.e. when $p = 1$). We conclude that `DIPS` significantly improves the data efficiency of the vanilla pseudo-labeling baselines, between **60-70%** more efficient for UPS and greedy-PL respectively, to reach the same level of performance.

**Takeaway.** We have demonstrated that data quantity is not the sole determinant of success in pseudo-labeling. We reduce the amount of data needed to achieve a desired test accuracy by leveraging the selection mechanism of `DIPS`. This highlights the significance of a multi-dimensional approach to pseudo-labeling, where a focus on quality reduces the data quantity requirements.

## 5.4 `DIPS` IMPROVES PERFORMANCE OF CROSS-COUNTRY PSEUDO-LABELING

**Goal.** To further assess the real-world benefit of `DIPS`, we consider the clinically relevant task of improving classifier performance using data from hospitals in different countries.

**Setup.** We assess a setup using Prostate cancer data from the UK (CUTRACT (PCUK, 2019)) to define $(\mathcal{D}_{\text{lab}}, \mathcal{D}_{test})$, which is augmented by $\mathcal{D}_{\text{unlab}}$, from US data (SEER (Duggan et al., 2016)). While coming from different countries, the datasets have interoperable features and the task is to predict prostate cancer mortality. We leverage the unlabeled data from the US to augment the small labeled dataset from the UK, to improve the classifier when used in the UK (on $\mathcal{D}_{test}$).

**Results.** Fig. 6 illustrates that greedy-PL and UPS benefit from DIPS's selection of labeled and pseudo-labeled samples, resulting in improved test performance. Hence, this result underscores that ignoring the labeled data whilst also naively selecting pseudo-labeled samples simply using confidence scores (as in greedy-PL) yields limited benefit. We provide further insights into the selection and gains by DIPS in Appendix C.

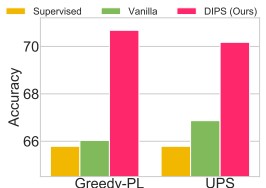

**Takeaway.** DIPS's selection mechanism improves performance when using semi-supervised approaches across countries.

Figure 6: Curation of $\mathcal{D}_{\text{lab}}$ permits us to better leverage a cross-country $\mathcal{D}_{\text{unlab}}$

### 5.5 DIPS WORKS WITH OTHER DATA MODALITIES

**Goal.** While DIPS is mainly geared towards the important problem of pseudo-labeling for tabular data, we explore an extension of DIPS to images, highlighting its versatility. Future work may explore other modalities like text or speech.

**Setup.** We investigate the use of DIPS to improve pseudo-labeling for CIFAR-10N (Wei et al., 2022a). With realism in mind, we specifically use this dataset as it reflects noise in image data stemming from real-world human annotations on M-Turk, rather than synthetic noise models (Wei et al., 2022a). We evaluate the semi-supervised algorithm FixMatch (Sohn et al., 2020) with a WideResNet-28 (Zagoruyko & Komodakis, 2016) for $n_{lab} = 1000$ over three seeds. FixMatch combines pseudo-labeling with consistency regularization, hence does not apply to the previous tabular data-focused experiments. We incorporate DIPS as a plug-in to the pseudo-labeling component of FixMatch.

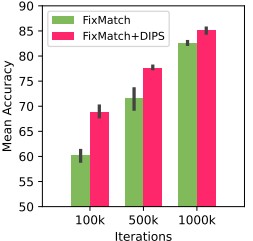

Figure 7: DIPS improves Fix-Match on CIFAR-10N

**Results.** Fig. 7 showcases the improved test accuracy for FixMatch+DIPS of **85.2%** over vanilla FixMatch of **82.6%**. A key reason is that DIPS discards harmful mislabeled samples from $\mathcal{D}_{\text{lab}}$, with example images shown in Fig. 8b. Furthermore, Table 8a shows the addition of DIPS improves time efficiency significantly, reducing the final computation time by **8 hours**. We show additional results for CIFAR-100N in Appendix C.4 and for other image datasets in Appendix C.4.4.

**Takeaway.** DIPS is a versatile framework that can be extended to various data modalities.

(a) DIPS improves the time efficiency (hours reported on a v100 GPU) of FixMatch, by 1.5-4X for the same performance. ↓ better

(b) Examples of mislabeled samples in CIFAR-10N discarded by DIPS. We note the incorrect labels and ideal ground-truth labels.

| Test acc (%) | FM + **DIPS** | FixMatch (FM) |
|---|---|---|
| 65 | **2.3** $_{\pm 0.4}$ | 8.0 $_{\pm 2.0}$ |
| 70 | **4.6** $_{\pm 0.5}$ | 16.4 $_{\pm 3.26}$ |
| 75 | **10.8** $_{\pm 0.7}$ | 26.5 $_{\pm 1.9}$ |
| 80 | **27.8** $_{\pm 0.8}$ | 35.8 $_{\pm 0.9}$ |
| 85 | **38.5** $_{\pm 0.3}$ | N.A. |

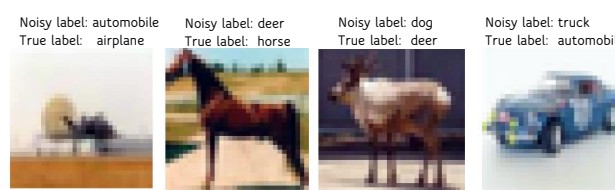

Figure 8: **(Left)** Time efficiency improvements **(Right)** Examples of harmful samples discarded

## 6 DISCUSSION

We propose DIPS, a plugin designed to improve *any* pseudo-labeling algorithm. DIPS builds on the key observation that the quality of labeled data is overlooked in pseudo-labeling approaches, while it is the core signal which renders pseudo-labeling possible. Motivated by real-world datasets and their inherent noisiness, we introduce a cleaning mechanism which operates both on labeled and pseudo-labeled data. We showed the value of taking into account labeled data quality – by characterizing and selecting data we improve test performance for various pseudo-labelers across 18 real-world datasets spanning tabular data and images. Future work could explore the impact of such insights for other modalities, including NLP and speech.

ETHICS AND REPRODUCIBILITY STATEMENTS

**Ethics Statement.** Improving machine learning with limited labels is a challenge with immense practical impact. It is especially important in high-stakes domains such as healthcare. We hope our paper encourages the community to reevaluate the "gold-standard" assumptions on the labeled dataset. It is unrealistic in reality and neglecting it can jeopardize the accuracy of classifiers trained with pseudo-labeling, which is highly problematic in settings such as healthcare. Note, in this work, we evaluate `DIPS` using multiple real-world datasets. The private datasets are de-identified and used in accordance with the guidance of the respective data providers.

**Reproducibility Statement.** Experiments are described in Section 5 with further details of the method and experimental setup/datasets included in Appendices A and B respectively. Code will be released upon acceptance.

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

# Appendix: Rethinking pseudo-labeling: Data-centric insights improve semi-supervised learning

## Table of Contents

# A    ADDITIONAL DETAILS ON DIPS

## A.1    DIPS ALGORITHM

We summarize how DIPS can be incorporated into *any* pseudo-labeling algorithm as per Algorithm 1. For clarity, we highlight in red how DIPS extends the current pseudo-labeling formulations.

---

**Algorithm 2** Plug DIPS into *any* pseudo-labeler

---

1: Train a network, $f^{(0)}$, using the samples from $\mathcal{D}_{\text{lab}}$.
2: Plug-in DIPS: set $\mathcal{D}_{\text{train}}^{(1)} = \mathcal{D}^{(1)} = r(\mathcal{D}_{\text{lab}}, f^{(0)})$          ▷ Initialization of the training set
3: **for** $t = 1..T$ **do**
4:     Initialize new network $f^{(t)}$
5:     Train $f^{(t)}$ using $\mathcal{D}_{\text{train}}^{(t)}$.
6:     Pseudo-label $\mathcal{D}_{\text{unlab}}$ using $f^{(t)}$
7:     Define $\mathcal{D}^{(t+1)}$ using the PL method's selector $s$
8:     Plug-in DIPS : Define $\mathcal{D}_{\text{train}}^{(t+1)} = r(\mathcal{D}^{(t+1)}, f^{(t)})$          ▷ Data characterization and selection, Sec. 4.3
9: **end for**
10: **return** $f_T$

---

We emphasize a key advantage of DIPS lies in its simplicity. Beyond getting the computation almost for free (no additional models to train when instantiating DIPS with learning dynamics) - i.e. **P3: Computationally cheap**, we can also plug DIPS into and augment *any* pseudo-labeling algorithm. (i.e. **P1: Plug & Play**), which makes for easier adoption.

## A.2    OVERVIEW OF PSEUDO-LABELING METHODS

As we described in Sec. 3.1, current pseudo-labeling methods typically differ in the way the selector function $s$ is defined. Note that $s$ is solely used to select *pseudo-labeled samples*, among those which have not already been pseudo-labeled at a previous iteration. The general way to define $s$ for any set $\mathcal{D}$ and function $f$ is $s(\mathcal{D}, f) = \{(x, [\hat{y}]_k) | x \in \mathcal{D}, \hat{y} = f(x), k \in [C], [m(x, f)]_k = 1\}$, where $m$ is such that $m(x, f) \in \mathbb{R}^C$. Alternatively stated, a selector $m$ outputs the binary decision of selecting $x$ and its associated pseudo-labels $\hat{y}$. Notice that we allow the multi-label setting, where $C > 1$, hence explaining why the selector $m$'s output is a vector of size $C$. We now give the intuition of how widely used PL methods construct the selector $m$, thus leading to specific definitions of $s$.

**Greedy pseudo-labeling (Lee et al., 2013)** The intuition of greedy pseudo-labeling is to select a pseudo-label if the classifier is sufficiently confident in it. Given two positive thresholds $\tau_p$ and $\tau_n$ with $\tau_n < \tau_p$, and a classifier $f$, $m$ is defined by $[m(x, f)]_k = \mathbb{1}([f(x)]_k \geq \tau_p) + \mathbb{1}([f(x)]_k \leq \tau_n)$ for $k \in [C]$.

**UPS (Rizve et al., 2021)** In addition to confidence, UPS considers the uncertainty of the prediction when selecting pseudo-labels. Given two thresholds $\kappa_n < \kappa_p$, UPS defines $[m(x, f)]_k = \mathbb{1}(u([f(x)]_k) \leq \kappa_p)\mathbb{1}([f(x)]_k \geq \tau_p) + \mathbb{1}(u([f(x)]_k) \leq \kappa_n)\mathbb{1}([f(x)]_k \leq \tau_n)$, where $u$ is a proxy for the uncertainty. One could compute $u$ using MC-Dropout (for neural networks) or ensembles.

**Flexmatch (Zhang et al., 2021)** FlexMatch dynamically adjusts a class-dependent threshold for the selection of pseudo-labels. The selection mechanism is defined by: $[m(x, f)]_k = \mathbb{1}(\max_j [f(x)]_j > \tau_t(\arg\max_i [f(x)]_i))$, i.e. at iteration $t$, an unlabeled point is selected if and only if the confidence of the most probable class is greater than its corresponding dynamic threshold.

**SLA (Tai et al., 2021) and CSA (Nguyen et al., 2022a)** The fundamental intuition behind SLA and CSA is to solve a linear program in order to assign pseudo-labels to unlabeled data, based on the predictions $f(x)$. The allocation of pseudo-labels considers both the rows (the unlabeled samples) and the columns (the classes), hence contrasts greedy pseudo-labeling, and incorporates linear constraints in the optimization problem. An approximate solution is found using the Sinkhorn-Knopp algorithm.

### A.3 LEARNING DYNAMICS PLOT

Key to our instantiation of `DIPS` is the analysis of learning dynamics. We illustrate in Fig. 9 for a pseudo-labeling run in Sec. 5.5 the learning dynamics of 6 individual samples. `DIPS` uses these dynamics to compute the metrics of confidence and aleatoric uncertainty, as explained in Sec. 4.2. This then characterizes the samples as *Useful* or *Harmful*.

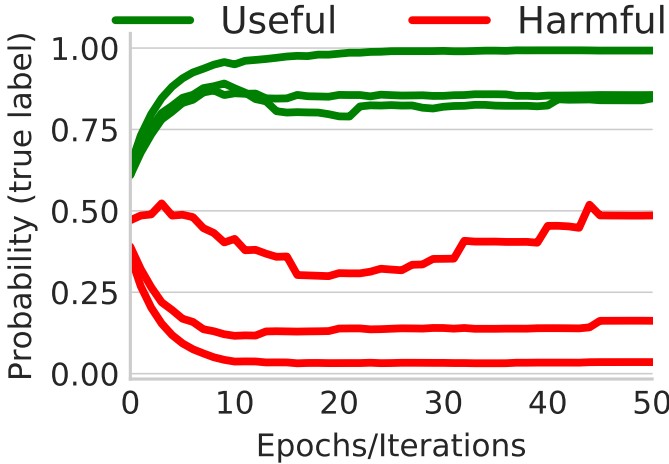

Figure 9: `DIPS` uses learning dynamics to compute the metrics of confidence and aleatoric uncertainty. It then characterizes labeled (and pseudo-labeled) samples based on these metrics. The $y$ axis represents the probability assigned to the true label by the current classifier $h$, which is $P(\hat{Y}_{\mathcal{F}}(x, y) = 1 | H = h)$.

The intuition is the following: a sample is deemed *Useful* if the classifiers at each checkpoint are confident in predicting the associated (pseudo-)label of the sample, and the associated aleatoric uncertainty is low. Failure to satisfy this criterion leads to a characterization as *Harmful*.

### A.4 COMPARISON WITH SCHMARJE ET AL. (2022)

In what follows, we compare DIPS with DC3 (Schmarje et al., 2022). While both DIPS and DC3 handle the data-centric issue of issues in data and share similarities in their titles, they tackle different data-centric problems which might arise in semi-supervised learning. The main differences are along 4 different dimensions:

1. **Problem setup/Type of data-centric issue**: DIPS tackles the problems of hard noisy labels where each sample has a single label assigned in the labeled set which could be incorrect. In contrast, DC3 deals with the problem of soft labeling where each sample might have multiple annotations from different annotators which may be variable.

2. **Label noise modeling**: DIPS aims to identify the noisy labels, whereas DC3 models the inter-annotator variability to estimate label ambiguity.

3. **Integration into SSL**: DIPS is a plug-in on top of any pseudo-labeling pipeline, selecting the labeled and pseudo-labeled data. DC3 on the otherhand uses its ambiguity model (learned on the multiple annotations) to either keep the pseudo-label or use a cluster assignment.

4. **Dataset applicability**: DIPS has lower dataset requirements as it can be applied to any dataset with labeled and unlabeled samples, even if there is only a single label per sample. It does not require multiple annotations. DC3 has higher dataset requirements as it relies on having multiple annotations per sample to estimate inter-annotator variability and label ambiguity. Without multiple labels per sample, it cannot estimate ambiguity and perform joint classification and clustering. Consequently, DIPS is applicable to the standard semi-supervised learning setup of limited labeled data and abundant unlabeled data, whereas DC3 targets the specific problem of ambiguity across multiple annotators.

## A.5 CONNECTION TO ACTIVE LEARNING

Active learning primarily focuses on the iterative process of selecting data samples that, when labeled, are expected to most significantly improve the model's performance. This selection is typically based on criteria such as uncertainty sampling which focuses on **epistemic uncertainty** (Mussmann & Liang, 2018; Houlsby et al., 2011; Kirsch et al., 2019; Nguyen et al., 2022b). The primary objective is to minimize labeling effort while maximizing the model's learning efficiency. In contrast, DIPS does both labeled and pseudo-labeled selection and employs the term 'useful' in a different sense. Here, 'usefulness' refers to the capacity of a data sample to contribute positively to the learning process based on its likelihood of being correctly labeled. Our approach, which leverages training dynamics based on **aleatoric uncertainty** and confidence, is designed to flag and exclude mislabeled data. This distinction is critical in our methodology as it directly addresses the challenge of data quality, particularly in scenarios where large volumes of unlabeled data are integrated into the training process. In active learning, these metrics are used to identify data points that, if labeled, would yield the most significant insights for model training. In our approach, they serve to identify and exclude data points that could potentially deteriorate the model's performance due to incorrect labeling.

# B EXPERIMENTAL DETAILS

## B.1 BASELINES

In our experiments, we consider the following baselines for pseudo-labeling: Greedy-PL (Lee et al., 2013), UPS (Rizve et al., 2021), Flexmatch (Zhang et al., 2021), SLA (Tai et al., 2021) and CSA (Nguyen et al., 2022a). To assess performance of `DIPS` for computer vision, we use FixMatch (Sohn et al., 2020).

**Greedy-PL** The confidence upper threshold is 0.8

**UPS** The confidence upper threshold is 0.8 and the threshold on the uncertainty is 0.2. The size of the ensemble is 10.

**FlexMatch** The upper threshold is 0.9 (which is then normalized).

**CSA and SLA** The confidence upper threshold is 0.8. The size of the ensemble is 20.

We use the implementation of these algorithms provided in (Nguyen et al., 2022a).

**FixMatch** The threshold is set to 0.95 as in (Sohn et al., 2020).

## B.2 DATASETS

We summarize the different datasets we use in this paper in Table 1. The datasets vary in number of samples, number of features and domain. Recall, we use data splits with a proportion of $\mathcal{D}_{\mathrm{lab}}$ : $\mathcal{D}_{\mathrm{unlab}}$ of 0.1:0.9.

Table 1: Summary of the datasets used

| Name | $n$ samples | $n$ features | Domain |
|------|------------|-------------|--------|
| Adult Income (Asuncion & Newman, 2007) | 30k | 12 | Finance |
| Agarius lepiota (Asuncion & Newman, 2007) | 8k | 22 | Agriculture |
| Blog (Buza, 2013) | 10k | 280 | Social media |
| Compas (Angwin et al., 2016) | 5k | 13 | Criminal justice |
| Covid-19 (Baqui et al., 2020) | 7k | 29 | Healthcare/Medicine |
| Credit (Taiwan) (Yeh & Lien, 2009) | 30k | 23 | Finance |
| CUTRACT Prostate (PCUK, 2019) | 2k | 12 | Healthcare/Medicine |
| Drug (Fehrman et al., 2017) | 2k | 27 | Healthcare/Medicine |
| German-credit (Asuncion & Newman, 2007) | 1k | 24 | Finance |
| Higgs (Baldi et al., 2014) | 25k | 23 | Physics |
| MAGGIC (Pocock et al., 2013) | 41k | 29 | Healthcare/Medicine |
| SEER Prostate (Duggan et al., 2016) | 20k | 12 | Healthcare/Medicine |

For computer vision experiments, we use CIFAR-10N (Wei et al., 2022a). The dataset can be accessed via its official release.[2]

## B.3 IMPLEMENTATION DETAILS

The three key design decisions necessary for pseudo-labeling are:

1. Choice of backbone model (i.e. predictive classifier $f$)
2. Number of pseudo-labeling iterations — recall that it is an iterative process by repeatedly augmenting the labeled data with selected samples from the unlabeled data.
3. Compute requirements.
   We describe each in the context of each experiment. For further details on the experimental setup and process, see each relevant section of the main paper.

---

[2]https://github.com/UCSC-REAL/cifar-10-100n

First, let's detail some general implementation details pertinent to all experiments.

■ Code: We will release code upon acceptance.

■ `DIPS` thresholds: Recall that `DIPS` has two thresholds $\tau_{\text{conf}}$ and $\tau_{\text{al}}$. We set $\tau_{\text{conf}} = 0.8$, in order to select high confidence samples based on the mean of the learning dynamic. Note, $\tau_{\text{al}}$ is bounded between [0,0.25]. We adopt an adaptive threshold for $\tau_{\text{al}}$ based on the dataset, such that $\tau_{\text{al}} = 0.75 \cdot (\max(v_{al}(\mathcal{D}_{\text{train}})) - \min(v_{al}(\mathcal{D}_{\text{train}})))$

### B.3.1 SYNTHETIC EXAMPLE: DATA CHARACTERIZATION AND UNLABELED DATA IMPROVE TEST ACCURACY

1. Backbone model: We use an XGBoost, with 100 estimators similar to (Nguyen et al., 2022a). Note, XGBoost has been shown to often outperform deep learning methods on tabular data (Borisov et al., 2021; Shwartz-Ziv & Armon, 2022). That said, our framework is not restricted to XGBoost.
2. Iterations: we use $T = 5$ pseudo-labeling iterations, as in (Nguyen et al., 2022a).
3. Compute: CPU on a MacBook Pro with an Intel Core i5 and 16GB RAM.

### B.3.2 TABULAR DATA EXPERIMENTS: SEC 5.2, 5.3, 5.4

1. Backbone model: Some of the datasets have limited numbers, hence we have the backbone as XGBoost, with 100 estimators similar to (Nguyen et al., 2022a). Note, XGBoost has been shown to often outperform deep learning methods on tabular data (Borisov et al., 2021; Shwartz-Ziv & Armon, 2022). That said, our framework is not restricted to XGBoost.
2. Iterations: we use T=5, pseudo-labeling iterations, as in (Nguyen et al., 2022a).
3. Compute: CPU on a MacBook Pro with an Intel Core i5 and 16GB RAM.

For the tabular datasets we use splits with a proportion of $\mathcal{D}_{\text{lab}} : \mathcal{D}_{\text{unlab}}$ of 0.1:0.9.

### B.3.3 COMPUTER VISION EXPERIMENTS: SEC 5.5

1. Backbone model: we use a WideResnet-18 (Zagoruyko & Komodakis, 2016) as in (Sohn et al., 2020).
2. Iterations: with Fixmatch we use T=1024k iterations as in (Sohn et al., 2020).
3. Data augmentations: Strong augmentation is done with RandAugment with random magnitude. Weak augmentation is done with a random horizontal flip.
4. Training hyperparameters: we use the same hyperparameters as in the original work (Sohn et al., 2020)
5. Compute: Nvidia V100 GPU, 6-Core Intel Xeon E5-2690 v4 with 16GB RAM.

For the experiments on CIFAR-10N, we use $\mathcal{D}_{\text{lab}} = 1000$ samples.

## C ADDITIONAL EXPERIMENTS

### C.1 ABLATION STUDY

**Goal.** We conduct an ablation study to characterize the importance of the different components of DIPS

**Experiment.** These components are:

- data characterization and selection for the **initialisation** of $\mathcal{D}^{(1)}$: DIPS defines $\mathcal{D}^{(1)} = \mathcal{D}_{\text{train}}^{(1)} = r(\mathcal{D}_{\text{lab}}, f^{(0)})$
- data characterization and selection at each **pseudo-labeling iteration**: DIPS defines $\mathcal{D}_{\text{train}}^{(i+1)} = r(\mathcal{D}^{(i+1)}, f^{(i)})$

Each of these two components can be ablated, resulting in four different possible combinations: **DIPS** (data selection both at initialization and during the pseudo-labeling iterations), **A1** (data selection only at initialization), **A2** (data selection only during the pseudo-labeling iterations), **A3** (no data selection). Note that **A3** corresponds to vanilla pseudo-labeling, and not selecting data amounts to using an identity selector.

We consider the same experimental setup as in Sec. 5.1, generating data in two quadrants with varying proportions of corrupted samples.

**Results.** The results are reported in Fig. 10. As we can see, each component in DIPS is important to improve pseudo-labeling: 1) the initialization of the labeled data $\mathcal{D}^{(1)}$ drives the pseudo-labeling process 2) data characterization of both labeled and pseudo-labeled samples is important.

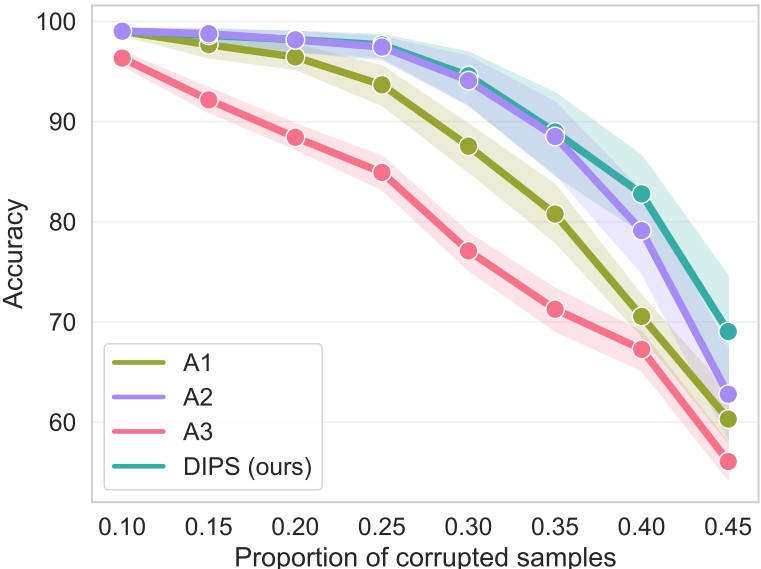

Figure 10: Ablation study: data characterization and selection is important both for the initialization of the labeled data and during the pseudo-labeling iterations, which forms the basis of DIPS.

**Takeaway.** It is important to characterize and select data both for the initialization of the labeled data and during the pseudo-labeling iterations.

## C.2 IMPACT OF THE SELECTOR FUNCTION $r$

**Goal.** We investigate variants of DIPS where we replace the selector function $r$ with heuristics used in the LNL literature. The most commonly used is the small-loss criterion (Xia et al., 2021). Additionally, we assess Fluctuation (Wei et al., 2022b) and FINE (Kim et al., 2021) as alternative sample selection approaches.

**Experiments.** We use the vanilla PL algorithm as the semi-supervised backbone. We consider three different selectors in addition to the one presented in the main paper:

- Small-loss criterion (Xia et al., 2021): the quantity of interest is $\mu(x,y) = \frac{1}{E} \sum_{e=1}^{E} l(x, y, f_e)$ where $l$ is a loss function and $f_1, ..., f_E$ are the classifiers at the different checkpoints. Intuitively, a sample with a high loss is more likely to present mislabeling issues, since it is harder to learn.
- Fluctuation criterion (Wei et al., 2022b): the quantity of interest, for two checkpoints $e_1 < e_2$ is $\beta(x, y, e_1, e_2) = 1([f_{e_1}(x)]_y > \frac{1}{2}) 1([f_{e_2}(x)]_y < \frac{1}{2})$, which is equal to one if the sample is correctly classified at the checkpoint $e_1$ and wrongly classified at $e_2$, 0 otherwise. Following (Wei et al., 2022b), we smooth this score with the confidence.
- FINE criterion (Kim et al., 2021): FINE creates a gram matrix of the representations in the noisy training dataset for each class. Then, FINE computes an alignment using the square of the inner product values between the representations and the first eigenvector of each gram matrix. A Gaussian mixture model is then fit on these alignment values to find clean and noisy instances.

The scores obtained by each approach are then used for sample selection, hence defining variants of the selector $r$.

**Results.** We report the results for 12 different datasets in Table 2. As we can see, the DIPS approach using learning dynamics outperforms the alternative LNL methods. This highlights the importance of a multi-dimensional data characterization, where both confidence and aleatoric uncertainty are taken into account to select samples. Moreover, the LNL methods typically operate under the assumption of a large number of labeled samples, highlighting that our DIPS approach tailored for the pseudo-labeling setting should indeed be preferred.

Table 2: DIPS outperforms heuristics used in the LNL setting by leveraging learning dynamics. Best performing method in **bold**, statistically equivalent performance underlined.

| | DIPS (OURS) | Small-Loss (Xia et al., 2021) | Fluctuation (Wei et al., 2022b) | FINE (Kim et al., 2021) |
|---|---|---|---|---|
| adult | **82.66 ± 0.10** | 79.52 ± 0.26 | 80.81 ± 0.20 | 24.22 ± 0.35 |
| agaricus-lepiota | **65.03 ± 0.25** | 64.45 ± 0.28 | 49.21 ± 1.48 | 35.96 ± 3.65 |
| blog | **80.58 ± 0.10** | 79.90 ± 0.28 | 80.05 ± 0.27 | 73.41 ± 1.66 |
| credit | **81.39 ± 0.07** | 78.46 ± 0.34 | 79.38 ± 0.29 | 64.58 ± 3.30 |
| covid | 69.97 ± 0.30 | **70.09 ± 0.36** | 69.03 ± 0.53 | 67.70 ± 0.84 |
| compas | **65.34 ± 0.25** | 62.76 ± 0.60 | 61.31 ± 0.64 | 60.20 ± 1.29 |
| cutract | **68.60 ± 0.31** | 66.33 ± 1.09 | 64.35 ± 1.49 | 61.98 ± 2.27 |
| drug | **78.16 ± 0.26** | 76.84 ± 0.61 | 75.66 ± 0.72 | 74.63 ± 1.98 |
| German-credit | 69.40 ± 0.46 | **69.80 ± 1.00** | 69.60 ± 1.19 | 69.70 ± 1.19 |
| higgs | **81.99 ± 0.07** | 81.08 ± 0.12 | 81.50 ± 0.09 | 73.82 ± 0.51 |
| maggic | **67.60 ± 0.08** | 65.57 ± 0.20 | 65.94 ± 0.20 | 62.28 ± 0.42 |
| seer | **82.74 ± 0.08** | 80.74 ± 0.26 | 82.28 ± 0.20 | 77.10 ± 0.76 |

**Takeaway.** A key component of DIPS is its sample selector based on learning dynamics, which outperforms methods designed for the LNL setting.

### C.3 INSIGHTS INTO DATA SELECTION

**Goal.** We wish to gain additional insight into the performance improvements provided by `DIPS` when added to the vanilla pseudo-labeling method. In particular, we examine the significant performance gain attained by `DIPS` for cross-country augmentation, shown in Section 5.4.

**Experiment.** The `DIPS` selector mechanism is the key differentiator as compared to the vanilla methods. Hence, we examine the samples selected samples from `DIPS` and vanilla. We then compare the samples to $\mathcal{D}_{\text{test}}$. Recall, we select samples from $\mathcal{D}_{\text{unlab}}$ which come from US patients, whereas $\mathcal{D}_{\text{lab}}$ and $\mathcal{D}_{\text{test}}$ are from the UK. We posit that "matching" the test distribution as closely as possible would lead to the best performance.

**Results.** We examine the most important features as determined by the XGBoost and compare their distributions. We find that the following 4 features in order are the most important: (1) Treatment: Primary hormone therapy, (2) PSA score, (3) Age, (4) Comorbidities. This is expected where the treatment and PSA blood scores are important predictors of prostate cancer mortality. We then compare these features in $\mathcal{D}_{\text{test}}$ vs the final $\mathcal{D}_{\text{train}}$ when using `DIPS` selection and using vanilla selection.

Fig. 11 shows that especially on the two most important features (1) Treatment: Primary hormone therapy and (2) PSA score; that `DIPS`'s selection better matches $\mathcal{D}_{\text{test}}$ — which explains the improved performance.

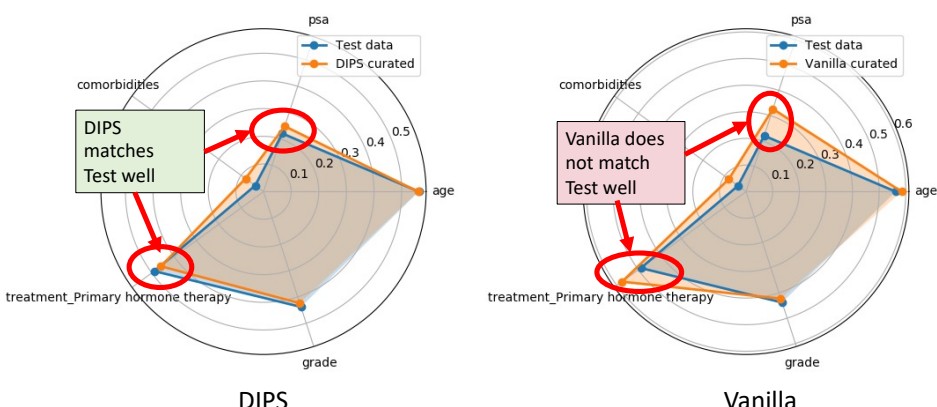

Figure 11: `DIPS` improves performance as its selection for $\mathcal{D}_{\text{train}}$ better matches $\mathcal{D}_{\text{test}}$ on the most important features, whereas vanilla selects samples which are different to $\mathcal{D}_{\text{test}}$

Quantitatively, we then compute the Jensen-Shannon (JS) divergence between the data selected by `DIPS` and vanilla as compared to $\mathcal{D}_{\text{test}}$. We find `DIPS` has a lower JSD of 0.0296 compared to vanilla of 0.0317, highlighting we better match the target domain's features through selection.

This behavior is reasonable, as samples that are very different will be filtered out by virtue of their learning dynamics being more uncertain.

For completeness, we also include a radar chart with vanilla PL but without any data selection in Fig. 12.

**Takeaway.** A significant source of `DIPS`'s gain is that its selection mechanism selects samples that closely approximate the distribution of $\mathcal{D}_{\text{test}}$. In particular, we see this holds true for the features which are considered most important to the classifier — hence accounting for the improved downstream model performance observed when using `DIPS`.

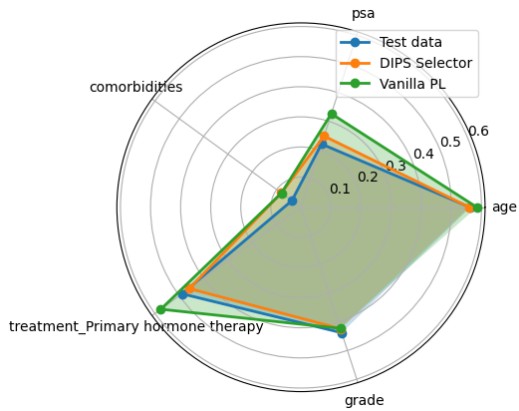

Figure 12: DIPS is closer to the test data than vanilla PL.

## C.4 ADDITIONAL EXPERIMENTS IN COMPUTER VISION

**Goal.** The goal of this experiment is to assess the benefit of `DIPS` in additional computer vision settings, namely:
(i) when increasing the number of classes to 100 (CIFAR-100N), with FixMatch
(ii) when the size of $\mathcal{D}_{\text{lab}}$ ($n_{\text{lab}}$) is small, with FixMatch and
(iii) when using a different pseudo-labeling algorithm — FreeMatch (Wang et al., 2023).

### C.4.1 DIPS IMPROVES THE PERFORMANCE OF FIXMATCH ON CIFAR-100N

**Goal.** The goal of this experiment is to further demonstrate `DIPS`'s utility in a setting with an increased number of classes.

**Setup.** We adopt the same setup as in Section 5.5, using the dataset CIFAR-100N (Wei et al., 2022a), which has 100 classes.

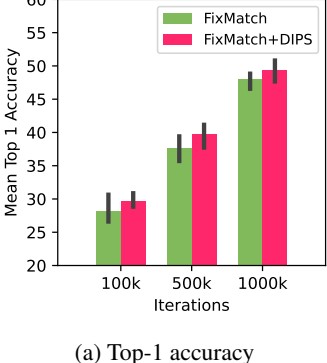
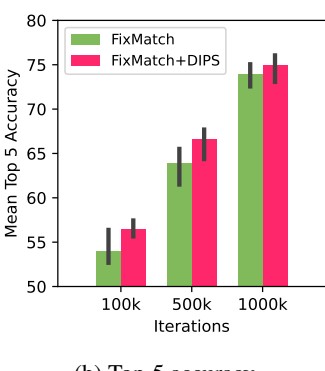

(a) Top-1 accuracy  (b) Top-5 accuracy

Figure 13: `DIPS` improves FixMatch on CIFAR-100N. We report both top-1 and top-5 accuracies.

**Results.** We show both top-1 and top-5 accuracies in Figure 13, for three different numbers of iterations and 3 different seeds, which highlight the performance gains obtained by using `DIPS` with FixMatch.

**Takeaway.** `DIPS` improves the performance of FixMatch on CIFAR-100N.

### C.4.2 DIPS IMPROVES FIXMATCH FOR SMALLER $n_{\text{lab}}$

**Setup.** We consider $n_{\text{lab}} = 200$, and use the same setup as in Section 5.5, considering the dataset CIFAR-10N.

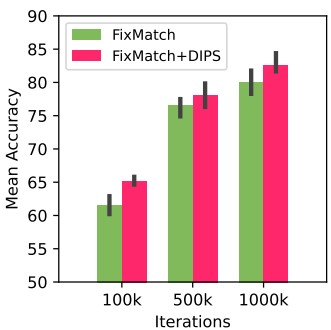

Figure 14: DIPS improves FixMatch with $n_{\text{lab}} = 200$ on CIFAR-10N

**Results.** We report the test accuracy in Figure 14 for three different numbers of iterations over 3 different seeds, highlighting the performance gains with DIPS.

**Takeaway.** DIPS improves performance of FixMatch for different sizes of $D_{\text{lab}}$

### C.4.3 DIPS IMPROVES THE PERFORMANCE OF FREEMATCH

**Goal.** The goal of this experiment is to further demonstrate DIPS's utility in computer vision with an alternative pseudo-labeling method, namely FreeMatch (Wang et al., 2023).

**Setup.** We adopt the same setup as in Section 5.5, using the dataset CIFAR-10N.

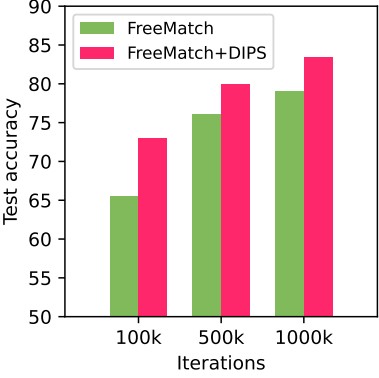

Figure 15: DIPS improves FreeMatch on CIFAR-10N

**Results.** We show the test accuracy in Figure 15, for three different numbers of iterations and highlight the performance gains by combining DIPS with FreeMatch to improve performance.

**Takeaway.** DIPS is versatile to other data modalities, improving the performance of FreeMatch on CIFAR-10N with the inclusion of DIPS.

### C.4.4 ADDITIONAL DATASETS

We present in Figure 16 results for 4 additional image datasets : satellite images from Eurosat Helber et al. (2019) and medical images from TissueMNIST which form part of the USB benchmark Wang et al. (2022). Additionally, we include the related OrganAMNIST, and PathMNIST, which are part of the MedMNIST collection Yang et al. (2021). Given that these datasets are well-curated, we consider a proportion of 0.2 of symmetric label noise added to these datasets. As is shown, DIPS consistently improves the FixMatch baseline, demonstrating its generalizability.

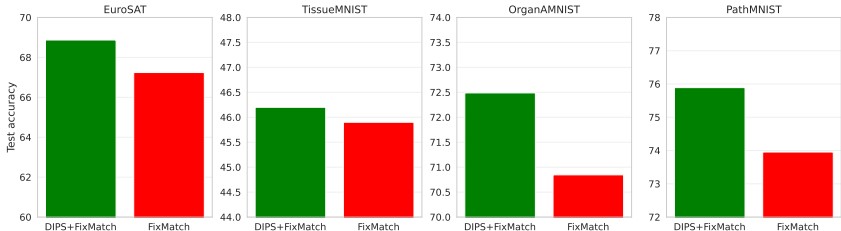

Figure 16: DIPS improves FixMatch on images datasets

## C.5 DEPENDENCY BETWEEN LABEL NOISE LEVEL AND AMOUNT OF LABELED DATA

We conduct a synthetic experiment following a similar setup as in Section 5.1 in our manuscript to investigate the dependency between label noise level and amount of labeled data. Note that the experiment is synthetic in order to be able to control the amount of label noise. We considered the same list of label noise proportions, ranging from 0. to 0.45. For each label noise proportion, we consider $n_{\text{lab}} \in \{50, 100, 1000\}$, and fix $n_{\text{unlab}} = 1000$. For each configuration we conduct the experiment 40 times. We report the results in Fig. 17. As we can see on the plots, PL+DIPS consistently outperforms the supervised baselines in almost all the configurations. When the amount of labeled data is low ($n_{lab} = 50$) and the proportion of corrupted samples is high (0.45), PL is on par with the supervised baseline. Hence pseudo-labeling is more difficult with a very low amount of labeled samples (and a high level of noise). We note, though, that DIPS consistently improves the PL baseline for reasonable amounts of label noise which we could expect in real-world settings (e.g. 0.1). The performance gap between DIPS and PL is remarkably noticeable for $n_{\text{lab}} = 1000$, i.e. when the amount of labeled samples is high.

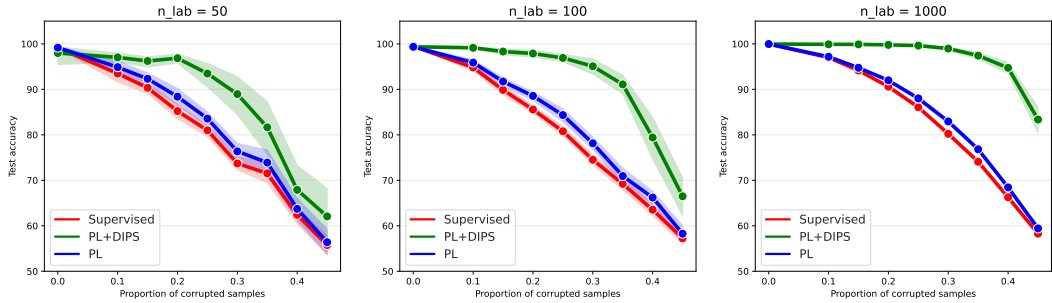

Figure 17: DIPS consistently improves upon the PL and supervised baselines, across the different label noise levels and amounts of labeled data.

## C.6 ABLATION ON THE TEACHER-STUDENT PSEUDO-LABELING FRAMEWORK

In Section 3.1, we decided to describe the common teacher-student pseudo-labeling methodology adopted in the tabular setting. As a consequence, we used the implementation provided by Nguyen et al. (2022a), which grows the training set with pseudo-labels generated at the current iteration, thus keeping old pseudo-labels in the training dataset in subsequent iterations. In addition to adopting this practice, we investigated this choice experimentally, by comparing between two versions of confidence-based pseudo-labeling:

- Version 1): with a growing set of pseudo-labels (as followed by the implementation of Nguyen et al. (2022a) and our paper)
- Version 2): without keeping old pseudo-labels.

We evaluate these two methods in the synthetic setup described in Section 5.2, and report the test accuracy in Fig. 18. The red line corresponds to Version 1) (the implementation we used in the manuscript), while the green line corresponds to Version 2). As we can see, in this tabular setting,

growing a training set by keeping the pseudo-labels generated at each iteration leads to the best results, motivating our adoption of this pseudo-labeling methodology used in the tabular setting.

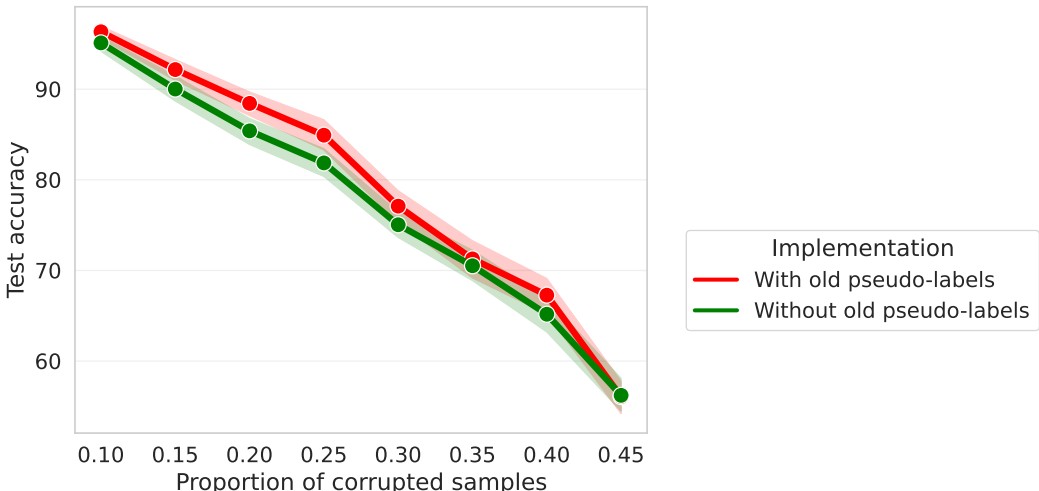

Figure 18: Growing the training set with the pseudo-labels generated at every iteration yields the best results

### C.7 ABLATION ON THE WINDOW OF ITERATIONS FOR LEARNING DYNAMICS

We conduct an experiment to investigate the choice of the range of iterations used to compute the learning dynamics. We consider ignoring the first 25%/50%/75% iterations, and use the remaining iterations to compute the learning dynamics. Figure 19 shows the mean performance difference by using the truncated iteration windows versus using all the iterations, and averages the results over the 12 datasets used in Section 5.2. As we can see, it is better to use all the iterations window, as the initial iterations carry some informative signal about the hardness of samples. This motivates our choice of computing the learning dynamics over the whole optimization trajectory, a choice which we adopt for all of our experiments.

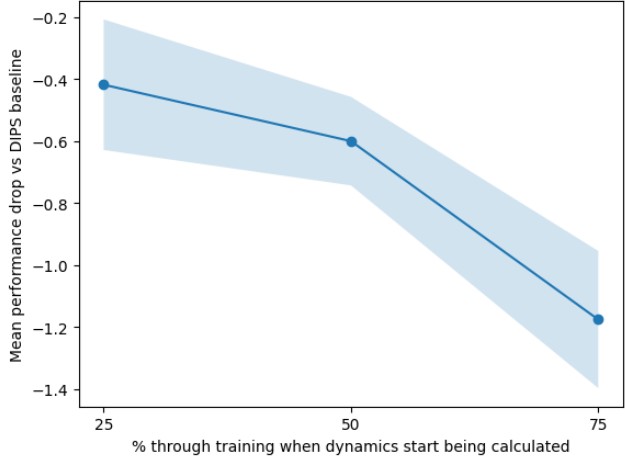

Figure 19: Computing the learning dynamics over the whole window of iterations yields the best results

## C.8    THRESHOLD CHOICES

We conduct an experiment in the synthetic setup where we vary the thresholds used for both the confidence and the aleatoric uncertainty. In addition to our choice used in our manuscript (confidence threshold = 0.8, and adaptive threshold on the aleatoric uncertainty), we consider two baselines:

- confidence threshold = 0.9 and uncertainty threshold = 0.1 (aggressive filtering)
- confidence threshold = 0.5 and uncertainty threshold = 0.2 (permissive filtering)

We show the test accuracy for these baselines in Figure 20. As we can see, our configuration achieves a good trade-off between an aggressive filtering configuration (red line) and a permissive one (blue line), which is why we adopt it for the rest of the experiments. We empirically notice in Section 5.2 that it performs well on the 12 real-world datasets we used.

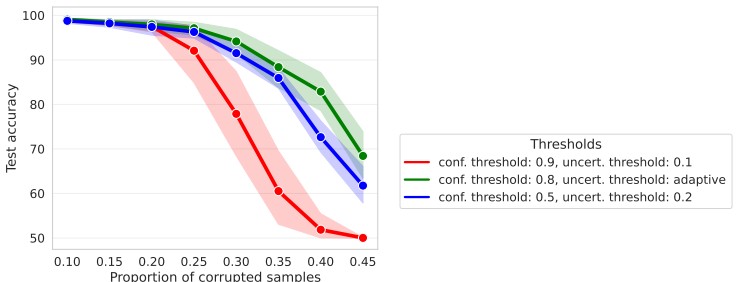

Figure 20: Our choice of thresholds performs better than an aggressive or a permissive filtering

## C.9    COMPARISON TO VIME

We compare DIPS with VIME (Yoon et al., 2020), by evaluating the two methods in the same setting as in Section 5.2. We report the results in Table 3. These results demonstrate that DIPS outperforms VIME across multiple real-world tabular datasets.

Table 3: DIPS outperforms VIME. Best performing method in **bold**, statistically equivalent performance underlined.

|  | DIPS (OURS) | VIME (Yoon et al., 2020) |
|---|---|---|
| adult | **82.66 ± 0.10** | 67.69 ± 0.10 |
| agaricus-lepiota | 65.03 ± 0.25 | **66.13 ± 0.01** |
| blog | **80.58 ± 0.10** | 73.52 ± 0.01 |
| credit | **81.39 ± 0.07** | 66.91 ± 0.02 |
| covid | **69.97 ± 0.30** | 68.28 ± 0.03 |
| compas | **65.34 ± 0.25** | 63.41 ± 0.02 |
| cutract | **68.60 ± 0.31** | 60.36 ± 0.04 |
| drug | **78.16 ± 0.26** | 74.47 ± 0.03 |
| German-credit | **69.40 ± 0.46** | 62.65 ± 0.05 |
| higgs | **81.99 ± 0.07** | 71.34 ± 0.03 |
| maggic | **67.60 ± 0.08** | 64.98 ± 0.01 |
| seer | **82.74 ± 0.08** | 80.12 ± 0.01 |

## C.10    IMPORTANCE OF ALEATORIC UNCERTAINTY

We conduct an ablation study where we remove the aleatoric uncertainty in `DIPS` and only keep a confidence-based selection (with threshold = 0.8). We term this confidence ablation to highlight if there is indeed value to the aleatoric uncertainty component of DIPS. We report results in Table 4, for

the 12 tabular datasets used in Section 5.2, which shows the benefit of the two-dimensional selection criterion of DIPS. Of course, in some cases there might not be a large difference with respect to our confidence ablation — however we see that DIPS provides a statistically significant improvement in most of the datasets. Hence, since the computation is negligible, it is reasonable to use the 2-D approach given the benefit obtained on the noisier datasets.

Table 4: Aleatoric uncertainty is a key component of DIPS

|  | DIPS | Confidence Ablation |
|---|---|---|
| adult | $\mathbf{82.66 \pm 0.10}$ | $82.13 \pm 0.16$ |
| agaricus-lepiota | $\mathbf{65.03 \pm 0.25}$ | $64.38 \pm 0.23$ |
| blog | $\mathbf{80.58 \pm 0.10}$ | $80.22 \pm 0.33$ |
| credit | $\mathbf{81.39 \pm 0.07}$ | $79.76 \pm 0.15$ |
| covid | $\mathbf{69.97 \pm 0.30}$ | $69.28 \pm 0.40$ |
| compas | $\mathbf{65.34 \pm 0.25}$ | $64.69 \pm 0.25$ |
| cutract | $\mathbf{68.60 \pm 0.31}$ | $66.32 \pm 0.12$ |
| drug | $\mathbf{78.16 \pm 0.26}$ | $75.37 \pm 0.71$ |
| higgs | $\mathbf{81.99 \pm 0.07}$ | $81.42 \pm 0.16$ |
| maggic | $\mathbf{67.60 \pm 0.08}$ | $66.26 \pm 0.18$ |
| seer | $\mathbf{82.74 \pm 0.08}$ | $82.02 \pm 0.15$ |

## D    BROADER IMPACT

In this work, we delve into the essential yet often neglected aspect of labeled data quality in the application of pseudo-labeling, a semi-supervised learning technique. Our key insights stem from a data-centric approach that underscores the role of 'labeled data quality' - a facet typically overlooked due to the default assumption of labeled data being 'perfect'. In stark contrast to the traditional, algorithm-centric pseudo-labeling literature which largely focuses on refining pseudo-labeling methods, we accentuate the critical influence of the quality of labeled data on the effectiveness of pseudo-labeling.

By way of introducing the `DIPS` framework, our work emphasizes the value of characterization and selection of labeled data, consequently improving any pseudo-labeling method. Moreover, akin to traditional machine learning problems, focusing on labeled data quality in the context of pseudo-labeling promises to lessen risks, costs, and potentially detrimental consequences of algorithm deployment. This perspective opens up many avenues for applications in areas where labeled data is scarce or expensive to acquire, including but not limited to healthcare, social sciences, autonomous vehicles, wildlife conservation, and climate modeling scenarios. Our work underscores the need for a data-centric paradigm shift in the pseudo-labeling landscape.

