# OpenReview forum: "Rethinking pseudo-labeling: Data-centric insights improve semi-supervised learning"
_ICLR.cc/2024/Conference — Submitted to ICLR 2024_

### Official Review · Reviewer_K9EF · 2023-10-28

**Soundness:** 3 good
**Presentation:** 3 good
**Contribution:** 3 good
**Rating:** 6
**Confidence:** 3

**Summary:**

This paper rethinks Semi-Supervised Learning from a data-centric view, that is, the labeled data may be not reliable and may contain noise in real-world applications. In this case, previous semi-supervised learning methods, which heavily rely on labeled data, are no longer applicable, showing low data efficiency. The author proposes a method for selecting high-quality data based on the average confidence and aleatoric uncertainty of historical predictions during the training process. Experimental results demonstrate that this method can identify reliable data, improve the data efficiency of semi-supervised learning, and adapt to different pseudo-labeling algorithms.

**Strengths:**

1. The proposal is simple and technologically reasonable.
2. The experiments seem comprehensive, and the author has analyzed the effectiveness and practicality of the proposed method from multiple perspectives. From the results, the proposed method has consistently achieved performance improvements.
3. Overall, this paper presents an interesting problem and provides a simple and effective solution. This could have a positive impact on the practical application of SSL in the real world.

**Weaknesses:**

1. This paper focuses more on tabular data, which is different from the image data that the main SSL algorithms currently focus on. The experiments conducted in this article regarding images are still limited.
2. This paper improves the robustness of SSL against noisy labeled data through a simple data selection method. However, further analysis is not provided regarding the reasons for the success of this data selection. Is there a theoretical connection between the statistical features of historical predictions (average confidence and aleatoric uncertainty in this paper) and the improvement in pseudo-label quality in SSL? Under what conditions does this method work effectively for the overall quality of labeled data, such as the proportion of label noise? More in-depth analysis can further improve this paper.

**Questions:**

Q: Can the author provide more discussion and analysis to demonstrate the conditions under which this proposal is successful?

---

> ### Author Response · Authors · 2023-11-16
> **Response to Reviewer K9EF**
>
> Dear Reviewer K9EF
>
> Thank you for your thoughtful comments and suggestions! We give answers to each of the following points in turn and highlight the updates to the revised manuscript. In addition, we have uploaded the revised manuscript. We hope this response alleviates your concerns, but please let us know if there are any remaining concerns.
>
> - A) Additional experiments in computer vision
> - B) Analysis of conditions of success
>
> ---
>
> ### (A) Additional experiments in computer vision
> We thank the reviewer for this point on applying DIPS to images datasets. To address this, we have conducted experiments on $4$ additional image datasets : satellite images from Eurosat [R1] and medical images from TissueMNIST which form part of the USB benchmark [R6]. Additionally, we include the related OrganAMNIST, and PathMNIST, which are part of the MedMNIST collection [R2].
> Given that these datasets are well-curated, we consider a proportion of $0.2$ of symmetric label noise added to these datasets.
>
> We report the results in the following figure available at: **https://i.imgur.com/NYmYGHs.png**
>
>
> As is shown, DIPS consistently improves the FixMatch baseline, demonstrating its generalizability beyond tabular datasets.
>
> Additionally, we wish to reiterate that beyond CIFAR-10N, we have also conducted additional experiments on CIFAR-100N in the Appendix.
>
> **UPDATE:** we have included these additional results in Appendix C.4.4 in our revised manuscript.
>
> ---
>
> ### (B) Analysis of conditions of success
> We thank the reviewer for the question on conditions of success of DIPS. To answer it, we conduct a synthetic experiment following a similar setup as in Sec. 5.1 in our manuscript. Our setup considers the dependencies between the amount of label noise and the amount of labeled data. Note that the experiment is synthetic in order to be able to control the amount of label noise.
> We considered the same list of label noise proportions as in Sec. 5.1, ranging from 0. to 0.45. For each label noise proportion, we consider $n_{\mathrm{lab}} \in$ {50,100,1000}, and fix $n_{\mathrm{unlab}} = 1000$.
> For each configuration, we conduct the experiment $40$ times.
> We report the results in the following plots available at: **https://i.imgur.com/rzFeFsl.png**
>
> As we can see on the plots, PL+DIPS consistently outperforms the supervised baselines and PL in almost all the configurations. The performance gap between DIPS and PL is remarkably noticeable for $n_{\mathrm{lab}}=1000$, that is, when the amount of labeled samples is high, which hints at the fact that curation is made easier with more samples, and hence leads to improved performance.
> When the amount of labeled data is very low ($n_{lab} = 50$) and the proportion of corrupted samples is high ($0.45$), we can see that the different baselines have closer accuracies. We note, though, that DIPS consistently improves the PL baseline for reasonable amounts of label noise which we could expect in real-world settings (e.g. 0.1).
>
> **UPDATE:** We have included these results in Appendix C.5 of the revised manuscript.
>
>
> The reviewer also asks about any "theoretical connection" involving "statistical features of historical predictions". Some previous works in the *supervised learning setting* have shown that learning dynamics emit a strong signal about the nature of samples ([R3],[R4]). For example, Theorem 3 in [R5] (Appendix A.4 in the corresponding paper) gives some intuition about mislabeled samples in the supervised setting, by stating that the probability that mislabeled examples are classified with their given (incorrect) labels tends to $0$ in a mixture setting, as the model is trained until convergence. As such, this phenomenon directly impacts the agreement between the predictions of the model and the given label of a mislabeled sample, the agreement being worse as the model converges. This agreement is directly the quantity that we capture with the learning dynamics (confidence and aleatoric uncertainty) and DIPS' selection mechanism $s$.
>
>
> We hope this answered your points, please let us know if there are any remaining concerns.

---

> ### Author Response · Authors · 2023-11-16
> **Response to Reviewer K9EF - References**
>
> ### References
>
> [R1] Patrick Helber, Benjamin Bischke, Andreas Dengel, and Damian Borth. Eurosat: A novel dataset and deep learning benchmark for land use and land cover classification. IEEE Journal of Selected Topics in Applied Earth Observations and Remote Sensing, 2019.
>
> [R2] Jiancheng Yang, Rui Shi, and Bingbing Ni. Medmnist classification decathlon: A lightweight automl benchmark for medical image analysis. In IEEE 18th International Symposium on Biomedical Imaging (ISBI), pages 191–195, 2021.
>
> [R3] Niladri S Chatterji and Philip M Long. Finite-sample analysis of interpolating linear classifiers in the overparameterized regime. J. Mach. Learn. Res., 22:129–1, 2021.
>
> [R4] Daniel Soudry, Elad Hoffer, Mor Shpigel Nacson, Suriya Gunasekar, and Nathan Srebro. The implicit bias of gradient descent on separable data. The Journal of Machine Learning Research, 19(1):2822–2878, 2018.
>
> [R5] Maini, Pratyush, Saurabh Garg, Zachary Lipton, and J. Zico Kolter. "Characterizing datapoints via second-split forgetting." Advances in Neural Information Processing Systems 35 (2022): 30044-30057.
>
> [R6] Wang, Yidong, et al. "Usb: A unified semi-supervised learning benchmark for classification." Advances in Neural Information Processing Systems 35 (2022): 3938-3961.

---

> ### Author Response · Authors · 2023-11-20
> **Dear Reviewer K9EF**
>
> Dear Reviewer K9EF
>
> We are sincerely grateful for your time and energy in the review process.We hope that our responses and appendix/manuscript updates have been helpful. Please let us know of any leftover concerns and if there was anything else we could do to address any further questions or comments!
>
> Thank you!
> Paper 3725 Authors

---

> > ### Comment · Reviewer_K9EF · 2023-11-20
> >
> > Thanks for your detailed response, which addressed my concerns. I will keep my score.

---

> ### Author Response · Authors · 2023-11-20
> **Thank you for your response**
>
> Thank you for your response, we are glad our additional experiments and analysis addressed your concerns. We would like to thank you for the encouragement to add further image datasets, which has improved the paper and more thoroughly illustrates the impact of DIPS and data quality across diverse settings --- which is now more extensive than other papers in this area (``see below Table``).
>
> Please let us know if there are any remaining points that we could address that would lead you to increase your score. We are eager to do our utmost to address them!
>
> --
> | Method              | Data Modalities and Number of Datasets                                 |
> |---------------------|----------------------------------------------------------------------|
> | __DIPS (Ours)__ | __Synthetic: 1, Tabular: 12, Images: 6__ |
> | Greedy PL [R4]          | Images: 1                  |
> | UPS     [R5]               | Images: 4                  |
> | FlexMatch   [R6]       |  Images: 5                  |
> |      SLA   [R7]                                 | Images: 3                  |
> | CSA [R8]      |  Tabular: 12, Images: 1  |
> | FixMatch  [R9] |  Images: 5  |
> |FreeMatch [R10] |  Images: 5  |
>
> ---
>
> ### References
>
> [R4] Dong-Hyun Lee et al. Pseudo-label: The simple and efficient semi-supervised learning method for deep neural networks. In Workshop on Challenges in Representation Learning, ICML, 2013.
>
> [R5] Mamshad Nayeem Rizve, Kevin Duarte, Yogesh S Rawat, and Mubarak Shah. In defense of pseudo-labeling: An uncertainty-aware pseudo-label selection framework for semi-supervised learning. In International Conference on Learning Representations, 2021.
>
> [R6] Bowen Zhang, Yidong Wang, Wenxin Hou, Hao Wu, Jindong Wang, Manabu Okumura, and Takahiro Shinozaki. FlexMatch: Boosting semi-supervised learning with curriculum pseudo labeling. Advances in Neural Information Processing Systems, 34, 2021
>
> [R7] Kai Sheng Tai, Peter D Bailis, and Gregory Valiant. Sinkhorn label allocation: Semi-supervised classification via annealed self-training. In International Conference on Machine Learning, pp. 10065–10075. PMLR, 2021.
>
> [R8] Vu-Linh Nguyen, Sachin Sudhakar Farfade, and Anton van den Hengel. Confident Sinkhorn allocation
> for pseudo-labeling. arXiv preprint arXiv:2206.05880, 2022a
>
> [R9]Kihyuk Sohn, David Berthelot, Nicholas Carlini, Zizhao Zhang, Han Zhang, Colin A Raffel, Ekin Dogus Cubuk, Alexey Kurakin, and Chun-Liang Li. FixMatch: Simplifying semi-supervised learning with consistency and confidence. Advances in Neural Information Processing Systems, 33, 2020.
>
> [R10]Yidong Wang, Hao Chen, Qiang Heng, Wenxin Hou, Yue Fan, Zhen Wu, Jindong Wang, Marios Savvides, Takahiro Shinozaki, Bhiksha Raj, Bernt Schiele, and Xing Xie. Freematch: Self-adaptive thresholding for semi-supervised learning. In The Eleventh International Conference on Learning
> Representations, 2023

---

### Official Review · Reviewer_hGhE · 2023-10-31

**Soundness:** 3 good
**Presentation:** 3 good
**Contribution:** 3 good
**Rating:** 5
**Confidence:** 4

**Summary:**

The paper addresses the challenge of dealing with inaccurately labeled data in semi-supervised learning contexts, where the small amount of labeled data available may contain errors. Contrary to the common presumption that labeled data is error-free, real-world scenarios often present labeling inaccuracies due to factors like mislabeling or ambiguity. To mitigate this problem, the authors introduce a method called DIPS, which is designed to discern and select the most reliable labeled data and pseudo-labels for unlabeled data throughout the training process.

**Strengths:**

1. The paper is grounded in a well-justified research gap, addressing the often overlooked errors in labeled data from prior studies.
2. The method proposed is both straightforward and impactful, demonstrating its efficacy across multiple tabular datasets as well as select small-scale computer vision datasets.
3. The authors have furnished extensive experimental details to facilitate reproduction.

**Weaknesses:**

1. While the authors assert that DIPS is intended to be a versatile tool that can be seamlessly merged with current pseudo-labeling strategies, the experiments are mainly limited to tabular datasets. It would be beneficial to extend testing to the USB[1] benchmark, encompassing datasets from computer vision, natural language processing, and speech to better demonstrate DIPS's generalizability. I will reconsider my score if more experiments are conducted.

2. The sections on Notation and Methodology are challenging to interpret. Transferring the pseudo-code to the Methodology section could enhance clarity and comprehension.

[1] Wang, Yidong, et al. "Usb: A unified semi-supervised learning benchmark for classification." Advances in Neural Information Processing Systems 35 (2022): 3938-3961.

**Questions:**

See weaknesses.

---

> ### Author Response · Authors · 2023-11-16
> **Response to Reviewer hGhE**
>
> Dear Reviewer hGhE
>
> Thank you for your thoughtful comments and suggestions! We give answers to each of the following points in turn and highlight the updates to the revised manuscript. In addition, we have uploaded the revised manuscript. We hope this response alleviates your concerns, but please let us know if there are any remaining concerns.
>
> - A) Additional experiments with the USB benchmark
> - B) Placement of pseudo-code
>
> ---
>
> ### (A) Additional experiments with the USB benchmark
> We thank the reviewer for this good suggestion to include datasets from the suggested USB benchmark, which, we believe, will strengthen our submission. To address this point, we have conducted experiments on $4$ additional datasets: satellite images from Eurosat [R1] and medical images from TissueMNIST which form part of the USB benchmark [R3]. Additionally, we include the related OrganAMNIST, and PathMNIST, which are part of the MedMNIST collection [R2].
> Given that these datasets are well-curated, we consider a proportion of $0.2$ of symmetric label noise added to these datasets.
>
> We report the results in the figure available at **https://i.imgur.com/NYmYGHs.png**
>
>
> As is shown, DIPS consistently improves the FixMatch baseline, demonstrating its generalizability beyond tabular datasets.
>
> Additionally, we wish to reiterate that beyond CIFAR-10N, we have also conducted additional experiments on CIFAR-100N in Appendix C.4.1.
>
> **UPDATE:** we have included these additional results in Appendix C.4.4 in our revised manuscript.
>
> ---
>
> ### (B) Placement of pseudo-code
> Thank you for your suggestion to improve the presentation of our work. We agree with you and have moved the pseudo-code presented in Algorithm 1 to Section 4.4 in the updated manuscript.
>
> **UPDATE:** Algorithm 1 moved to Section 4.4 in the revised manuscript.
>
> We hope this answers your points, please let us know if there are any remaining concerns.
>
> ---
>
> ### References
>
> [R1] Patrick Helber, Benjamin Bischke, Andreas Dengel, and Damian Borth. Eurosat: A novel dataset and deep learning benchmark for land use and land cover classification. IEEE Journal of Selected Topics in Applied Earth Observations and Remote Sensing, 2019.
>
> [R2] Jiancheng Yang, Rui Shi, and Bingbing Ni. Medmnist classification decathlon: A lightweight automl benchmark for medical image analysis. In IEEE 18th International Symposium on Biomedical Imaging (ISBI), pages 191–195, 2021.
>
> [R3]  Wang, Yidong, et al. “Usb: A unified semi-supervised learning benchmark for classification.” Advances in Neural Information Processing Systems 35 (2022): 3938-3961.

---

> ### Author Response · Authors · 2023-11-20
> **Dear Reviewer hGhE**
>
> Dear Reviewer hGhE
>
> We are sincerely grateful for your time and energy in the review process.We hope that our responses and appendix/manuscript updates have been helpful. Please let us know of any leftover concerns and if there was anything else we could do to address any further questions or comments!
>
> Thank you!
> Paper 3725 Authors

---

> > ### Comment · Reviewer_hGhE · 2023-11-20
> >
> > Thank you for your response. I appreciate the provided experiments, however, I believe further experiments, particularly with semi-aves and STL-10 datasets, would significantly strengthen the evaluation of DIPS. Additionally, my query regarding the performance of DIPS on natural language and speech datasets remains unaddressed. Understanding its effectiveness in these domains is crucial for a comprehensive assessment. Due to these limitations in the scope of your experiments, I am inclined to maintain my original evaluation score.

---

> ### Author Response · Authors · 2023-11-20
> **Thank you for your response**
>
> Dear Reviewer hGhE,
>
> Thank you once again for your insightful feedback and for acknowledging the new experiments we have added to our manuscript. We would like to take this opportunity to make the purpose of our paper clearer, which is as follows:
>
> - We identify the issue of data quality and mislabeling in the original labeled dataset and highlight the importance of this for pseudo-labeling.
> - We propose a general framework to incorporate this insight within the pseudo-labeling paradigm and introduce an approach, DIPS, that evaluates the quality of the labeling using model confidence and aleatoric uncertainty.
> - We demonstrate the application and impact of this approach in both the tabular and image modalities.
>
> Your review has been very helpful for us in disentangling these three points and we have clarified them as a result in the manuscript.
>
> In short, we don’t use experiments to demonstrate the universal superiority of our proposed approach, DIPS (point 2). Rather we seek to show the usefulness and applicability of our key insight (point 1 above) by applying DIPS to real-world cases (point 3 above), and in particular those where manual data audit might be particularly hard (e.g. tabular). We believe others will find this approach useful too and they will be able to use it and develop new approaches in other modalities, such as those you suggest.
>
> So, our paper identifies a fundamental issue that has been overlooked previously in semi-supervised learning, introduces a new method to tackle this limitation, and then shows its usefulness with some real-world examples.
>
> We hope this clarifies the goal of our paper and we have **updated the paper** (updates in purple) to reflect the clarity that your review has helped us to reach.

---

> ### Author Response · Authors · 2023-11-20
> **Re:our experiments**
>
> Regarding our evaluation, we believe that the scope of our experiments is substantially broad to achieve our stated aims in the previous comment. We have demonstrated the impact and pervasiveness of the identified issue across multiple modalities and 19 datasets. This is substantially higher than the median of 4 datasets (mean: 4.1) employed to evaluate methods published at leading machine learning conferences (Liao et al. [R11]).
>
> We wish to draw the Reviewer's attention to the validation of other approaches. We checked the modalities and number of datasets employed by the baseline approaches used in our manuscript, which we have included in the table below. We confirmed our experiments were more extensive than any other approach, both in terms of the number of datasets and modalities. CSA is the only other approach assessed on non-image data, and no approach conducted experiments on NLP or speech data.
>
> We agree that NLP and speech are interesting domains for future research building on the formalism and problem introduced by DIPS, and we have updated Section 5 of our manuscript accordingly. However, as noted above, they are not the central focus of our work, which was to demonstrate that explicitly accounting for labeled data quality is an aspect that helps pseudo-labeling.
>
> We trust that this, together with clarifying changes included in our revised manuscript, allays any concerns the Reviewer had regarding the thoroughness of our evaluation of DIPS.
>
> | Method              | Data Modalities and Number of Datasets                                 |
> |---------------------|----------------------------------------------------------------------|
> | Greedy PL [R4]          | Images: 1                  |
> | UPS     [R5]               | Images: 4                  |
> | FlexMatch   [R6]       |  Images: 5                  |
> |      SLA   [R7]                                 | Images: 3       |
> | CSA [R8]      |  Tabular: 12, Images: 1  |
> | FixMatch  [R9] |  Images: 5  |
> |FreeMatch [R10] |  Images: 5 |
> | __DIPS (Ours)__ | __Synthetic: 1, Tabular: 12, Images: 6__ |
>
>
> ### References
> [R4] Dong-Hyun Lee et al. Pseudo-label: The simple and efficient semi-supervised learning method for deep neural networks. In Workshop on Challenges in Representation Learning, ICML, 2013.
>
> [R5] Mamshad Nayeem Rizve, Kevin Duarte, Yogesh S Rawat, and Mubarak Shah. In defense of pseudo-labeling: An uncertainty-aware pseudo-label selection framework for semi-supervised learning. In International Conference on Learning Representations, 2021.
>
> [R6] Bowen Zhang, Yidong Wang, Wenxin Hou, Hao Wu, Jindong Wang, Manabu Okumura, and Takahiro Shinozaki. FlexMatch: Boosting semi-supervised learning with curriculum pseudo labeling. Advances in Neural Information Processing Systems, 34, 2021
>
> [R7] Kai Sheng Tai, Peter D Bailis, and Gregory Valiant. Sinkhorn label allocation: Semi-supervised classification via annealed self-training. In International Conference on Machine Learning, pp. 10065–10075. PMLR, 2021.
>
> [R8] Vu-Linh Nguyen, Sachin Sudhakar Farfade, and Anton van den Hengel. Confident Sinkhorn allocation
> for pseudo-labeling. arXiv preprint arXiv:2206.05880, 2022a
>
> [R9]Kihyuk Sohn, David Berthelot, Nicholas Carlini, Zizhao Zhang, Han Zhang, Colin A Raffel, Ekin Dogus Cubuk, Alexey Kurakin, and Chun-Liang Li. FixMatch: Simplifying semi-supervised learning with consistency and confidence. Advances in Neural Information Processing Systems, 33, 2020.
>
> [R10]Yidong Wang, Hao Chen, Qiang Heng, Wenxin Hou, Yue Fan, Zhen Wu, Jindong Wang, Marios Savvides, Takahiro Shinozaki, Bhiksha Raj, Bernt Schiele, and Xing Xie. Freematch: Self-adaptive thresholding for semi-supervised learning. In The Eleventh International Conference on Learning
> Representations, 2023
>
> [R11] Thomas Liao, Rohan Taori, Deborah Raji, and Ludwig Schmidt. Are We Learning Yet? A Meta Review of Evaluation Failures Across Machine Learning. NeurIPS Track on Datasets and Benchmarks, 2021

---

### Official Review · Reviewer_hxp2 · 2023-10-31

**Soundness:** 3 good
**Presentation:** 4 excellent
**Contribution:** 2 fair
**Rating:** 6
**Confidence:** 4

**Summary:**

This paper explores an often overlooked scenario in pseudo-labels (PLs) in semi-supervised learning (SSL), where the labeled data used for training is considered perfect. This paper breaks this assumption and shows that noise in the initial labeled set can be propagated to the pseudo-labels and hurt the final performance. To tackle this issue, a Data-centric Insights for semi-supervised learning (DIPS) framework is proposed. It selectively uses useful training samples among labeled and pseudo-labeled examples for every self-training iterations based on the learning dynamics of each example. DIPs has three properties which can be practical in real use and experiments are conducted on various real world datasets across different modalities.

**Strengths:**

- The motivation for considering the quality of labeled data is clearly presented, connected to real use cases. In addition, the pilot experiment in Figure 3 demonstrates that addressing the inherent noise in labeled data is necessary and that previous (standard) pseudo-labeling algorithms can fail on this setting.

- Writing quality needs to be acknowledged. All sections are well organized, and easy to follow. Especially for the experiments part, dividing a section into several paragraphs and giving a short summary of the results was a good idea. In addition a supplementary material covers a lot of details including additional ablation experiments.

- Although a simple data filtering method, the effectiveness of the proposed DIPS was impressive, since it achieves consistent improvements on various real-world cases.

**Weaknesses:**

- It seems that this paper is not the first to concern the inherent label noise in the labeled data and take the data-centric approach. It is necessary to discuss and compare with [L. Schmarje et al.,2022] to more clarify the conceptual novelty.

- Connection to the active learning literature is missing. Selection metric for the ‘useful’ data samples among the unlabeled data pool is of central interest in active learning. The term ‘usefulness’ can include various criterions, such as confidence and uncertainty (as in this work), coverage, diversity, and etc. In that sense, more comprehensive discussions and comparisons on selection metrics are expected. Related to this issue, how the quality and diversity (e.g., class distributions) of the selected training samples change for every generation?

- Choice of the threshold parameters $\tau_{\text{conf}}$ and $\tau_{\text{al}}$ seems to be very important. For example, highly tight thresholds (i.e., high $\tau_{\text{conf}}$ and low $\tau_{\text{al}}$) will remain only a few samples for training likely to be correct and abundant samples yet include noise for the vice versa. As the proposed algorithm is not designed to make a correction on the mislabeled samples but filter potentially harmful examples, exploring such trade-offs between remaining data proportion and performance depending on the thresholds will be valuable.

- Considering the computational aspects, it could be nearly free when the scale of unlabeled data is relatively small (i.e., in vision domain, CIFAR-10/100). In other words, when the scale of the unlabeled dataset grows large (i.e., million-scale samples such as ImageNet), computational overheads caused by evaluating labeled and pseudo-labeled examples with all checkpoints in previous rounds cannot be ignored.

- Although the presented work mainly targets semi-supervised learning on tabular data, baselines have been taken only from the image domain and comparison with SSL methods specific to tabular data such as [J. Yoon et al., 2020] is not given.

**Questions:**

- The proposed DIPS framework follows the iteration-based self-training scheme, while a typical pseudo-label-based SSL algorithm such as FixMatch doesn’t. FixMatch takes both labeled and unlabeled data and learns from both supervision signals (i.e., labels and online pseudo-labels). But the self-training scheme makes offline pseudo-labels after an iteration phase and some selected pseudo-labeled examples considered labeled data in the self-training next iteration. It seems to be a conflict between two different mechanisms. Hence, an illustrative example of how the non-iteration-based SSL algorithms can be incorporated into the DIPS framework is expected.

---

References

[L. Schmarje et al.,2022] A data-centric approach for improving ambiguous labels with combined semi-supervised classification and clustering, in ECCV 2022.

[J. Yoon et al., 2020] VIME: Extending the Success of Self- and Semi-supervised Learning to Tabular Domain, in NeurIPS 2020.

---

> ### Author Response · Authors · 2023-11-16
> **Response to Reviewer hxp2 [Part 1/3]**
>
> Dear Reviewer hxp2
>
> Thank you for your thoughtful comments and suggestions!
>
> We give answers to each of the following points in turn and highlight the updates to the revised manuscript. In addition, we have uploaded the revised manuscript. We hope this response alleviates your concerns, but please let us know if there are any remaining concerns.
>
> - A) Contrasting DIPS w/ Schmarje et al.,2022 **[Part 2/3]**
> - B) Connection to active learning **[Part 2/3]**
> - C) Threshold parameters & trade-offs **[Part 2/3]**
> - D) Computational trade-offs **[Part 3 /3]**
> - E) VIME baseline **[Part 3/3]**
> - F) Incorporating DIPS w/ Fixmatch **[Part 3/3]**

---

> ### Author Response · Authors · 2023-11-16
> **Response to Reviewer hxp2 [Part 2/3]**
>
> ### (A) Contrasting DIPS w/ Schmarje et al.,2022 [DC3]
>
>
> Thank you for highlighting the paper [L. Schmarje et al.,2022]. While both DIPS and DC3 handle the data-centric issue of issues in data and share similarities in their titles, they tackle different data-centric problems which might arise in semi-supervised learning. The main differences are along 4 different dimensions.
>
> (i) **Problem setup/Type of data-centric issue**: DIPS tackles the problems of hard noisy labels where each sample has a single label assigned in the labeled set which could be incorrect. In contrast, DC3 deals with the problem of soft labeling where each sample might have multiple annotations from different annotators which may be variable.
>
> (ii) **Label noise modeling**: DIPS aims to identify the noisy labels, whereas DC3 models the inter-annotator variability to estimate label ambiguity.
>
> (iii) **Integration into SSL**: DIPS is a plug-in on top of any pseudo-labeling pipeline, selecting the labeled and pseudo-labeled data. DC3 on the other hand uses its ambiguity model (learned on the multiple annotations) to either keep the pseudo-label or use a cluster assignment.
>
> (iv) **Dataset applicability**: DIPS has lower dataset requirements as it can be applied to any dataset with labeled and unlabeled samples, even if there is only a single label per sample. It does not require multiple annotations. DC3 has higher dataset requirements as it relies on having multiple annotations per sample to estimate inter-annotator variability and label ambiguity. Without multiple labels per sample, it cannot estimate ambiguity and perform joint classification and clustering. Consequently, DIPS is applicable to the standard semi-supervised learning setup of limited labeled data and abundant unlabeled data, whereas DC3 targets the specific problem of ambiguity across multiple annotators.
>
> **UPDATE:** we have added this discussion in realtion to Schmarje et al.,2022 to Appendix A.4 in our revised manuscript.
>
> ---
>
> ### (B) Connection to active learning
> We wish to clarify the difference in setting of DIPS vs active learning. We agree that the concept of 'usefulness' in the selection of data samples is a significant aspect of both active learning and our work. However, it is crucial to highlight the distinct contexts in which this term is used in both settings.
>
> Active learning primarily focuses on the iterative process of selecting data samples that, when labeled, are expected to most significantly improve the model's performance. This selection is typically based on criteria such as uncertainty sampling which focuses on **epistemic uncertainty** [R1-R4]. The primary objective is to minimize labeling effort while maximizing the model's learning efficiency.
>
> In contrast, DIPS does both labeled and pseudo-labeled selection and employs the term 'useful' in a different sense. Here, 'usefulness' refers to the capacity of a data sample to contribute positively to the learning process based on its likelihood of being correctly labeled. Our approach, which leverages training dynamics based on **aleatoric uncertainty** and confidence, is designed to flag and exclude mislabeled data. This distinction is critical in our methodology as it directly addresses the challenge of data quality, particularly in scenarios where large volumes of unlabeled data are integrated into the training process.
>
> In active learning, these metrics are used to identify data points that, if labeled, would yield the most significant insights for model training. In our approach, they serve to identify and exclude data points that could potentially deteriorate the model's performance due to incorrect labeling.
>
> **UPDATE:** We have included this discussion in Appendix A.5 in our revised manuscript.
>
> ---
>
> ### (C) Threshold parameters & trade-offs
>
> We thank the reviewer for the question on the choice of threshold parameters. To address this point, we conducted an experiment in the synthetic setup where we varied the thresholds used for both the confidence and the aleatoric uncertainty. In addition to our choice used in our manuscript (confidence threshold = 0.8, and adaptive threshold on the aleatoric uncertainty), we consider two baselines:
> 1) confidence threshold = 0.9 and uncertainty threshold = 0.1 (aggressive filtering)
> 2) confidence threshold = 0.5 and uncertainty threshold = 0.2 (permissive filtering)
>
> We show the test accuracy for these baselines in the following plot available at: https://i.imgur.com/QCYEgzm.png.
>
> As we can see, our configuration outperforms both the aggressive filtering configuration (red line) and a permissive one (blue line), which is why we adopt it for the rest of the experiments. We empirically notice in Section 5.2 that it performs well on the 12 real-world datasets we used.
>
> **UPDATE:** we have included these experimental results in Appendix C.8 in our revised manuscript.

---

> ### Author Response · Authors · 2023-11-16
> **Response to Reviewer hxp2 [Part 3/3]**
>
> ### (D) Computational trade-offs
> Thank you for asking a question about the computational overheads in our method, particularly in relation to large-scale datasets. While evaluating labeled and pseudo-labeled examples to compute the learning dynamics might incur additional overhead, we want to highlight that our selection mechanism yields a direct improvement in the quality of data used for training. This effect leads to a quicker model convergence. We refer to Fig. 8) a) in our manuscript, which gives the time efficiency of DIPS for the dataset CIFAR-10N. It shows that DIPS accelerates convergence by a factor 1.5-4X, despite the computation of learning dynamics.
>
> ---
>
> ### (E) VIME baseline
>
> We would like to thank the reviewer for suggesting a comparison to VIME. We believe its incorporation has helped to strengthen the paper. We report the results of VIME tested on the same setup as DIPS in the next table, where we use the same 12 datasets as in Section 5.2. These results demonstrate that DIPS outperforms VIME across multiple real-world tabular datasets.
>
> |                   | DIPS (OURS)         | VIME |
> |-------------------|---------------------|---------------------|
> | adult             | **82.66 ± 0.10**    | 67.69 ± 0.10        |
> | agaricus-lepiota  | 65.03 ± 0.25        | **66.13 ± 0.01**    |
> | blog              | **80.58 ± 0.10**    | 73.52 ± 0.01        |
> | credit            | **81.39 ± 0.07**    | 66.91 ± 0.02        |
> | covid             | **69.97 ± 0.30**    | 68.28 ± 0.03        |
> | compas            | **65.34 ± 0.25**    | 63.41 ± 0.02        |
> | cutract           | **68.60 ± 0.31**    | 60.36 ± 0.04        |
> | drug              | **78.16 ± 0.26**    | 74.47 ± 0.03        |
> | German-credit     | **69.40 ± 0.46**    | 62.65 ± 0.05        |
> | higgs             | **81.99 ± 0.07**    | 71.34 ± 0.03        |
> | maggic            | **67.60 ± 0.08**    | 64.98 ± 0.01        |
> | seer              | **82.74 ± 0.08**    | 80.12 ± 0.01        |
>
> **UPDATE:** we have included these experimental results in Appendix C.9 in our revised manuscript.
>
> ---
>
> ### (F) Incorporating DIPS w/ non-iteration-based SSL
> We thank the reviewer for asking this question on the integration of DIPS in non-iteration-based SSL.
> The DIPS framework necessitates two ingredients: a list of model checkpoints, and a label for each sample for which we want to compute the learning dynamics. In the case of non-iteration-based SSL methods, DIPS updates the learning dynamics at every training step and performs its selection of labeled and unlabeled data every $k$ training steps (for example, a training step could denote an epoch). For the unlabeled data, DIPS uses the pseudo-labels computed by the model at the beginning of every learning dynamics cycle. Then, after $k$ training steps, DIPS curates the labeled and unlabeled data using its selector function and then updates the pseudo-labels used for the computation of learning dynamics on unlabeled data for the next $k$ training steps.
>
> ---
>
> We hope this answers your points, please let us know if there are any remaining concerns.
>
> ---
>
> ### References
>
>
> [R1] Stephen Mussmann and Percy Liang. On the relationship between data efficiency and error for uncertainty sampling, 35th International Conference on Machine Learning, PMLR.
>
> [R2]Neil Houlsby, Ferenc Huszar, Zoubin Ghahramani, and Mate Lengyel. Bayesian active learning for classification and preference learning. arXiv preprint arXiv:1112.5745, 2011.
>
> [R3] Andreas Kirsch, Joost van Amersfoort, and Yarin Gal. Batchbald: Efficient and diverse batch acquisition for deep bayesian active learning. In Advances in Neural Information Processing Systems, pp. 7024–7035, 2019.
>
> [R4] Nguyen, Vu-Linh, Mohammad Hossein Shaker, and Eyke Hüllermeier. "How to measure uncertainty in uncertainty sampling for active learning." Machine Learning 111, no. 1 (2022): 89-122.

---

> ### Author Response · Authors · 2023-11-20
> **Dear Reviewer hxp2**
>
> Dear Reviewer hxp2
>
> We are sincerely grateful for your time and energy in the review process.We hope that our responses and appendix/manuscript updates have been helpful. Please let us know of any leftover concerns and if there was anything else we could do to address any further questions or comments!
>
> Thank you!
> Paper 3725 Authors

---

> ### Comment · Reviewer_hxp2 · 2023-11-22
> **Comments on the rebuttal**
>
> Thank you for your comprehensive comments, including detailed clarifications, discussions, and follow-up experiments. I have also carefully checked the discussions with other reviewers. I have a few additional comments as following.
>
> **(A) Comparison with DIPS and Schmarje et al., 2022 [DC3]**
>
> The conceptual comparison with existing work effectively highlights the uniqueness of the DIPS framework. I recommend extending this comparison to the other work [X. Wang et al., 2022], which also focuses on selective labeling. A broader comparison would further enhance the conceptual novelty of the proposed work.
>
>
> **(E) VIME Baseline**
>
> The experimental comparison with the Tabular-specific SSL method is appreciated. For a more comprehensive understanding, could you also include the numerical results of the supervised baseline and the vanilla PL method in the table? In addition, it would be beneficial to incorporate additional SSL methods specific to the tabular domain. While time constraints may limit this during the review period, please consider addressing this in your post-review revisions.
>
>
> **(F) Incorporation of DIPS with Non-Iteration-Based SSL**
>
> There seems to be a fundamental difference between DIPS’s approach and continuous training mechanisms like FixMatch, which generate instant pseudo-labels within the same training step. This distinction in the application and updating of pseudo-labels requires further clarification.It would be important to understand whether DIPS, when integrated with FixMatch, utilizes two distinct types of pseudo-labels: those derived from self-training and those from continuous training, similar to the way with [C. Wei et al., 2022]. A more explicit explanation of how DIPS can integrate with continuous training SSL methods, such as FixMatch, would be beneficial.
>
> —
>
> References
>
> [X. Wang et al., 2022] Unsupervised Selective Labeling for More Effective Semi-Supervised Learning, in ECCV 2022.
>
> [C. Wei et al., 2022] CReST: A Class-Rebalancing Self-Training Framework for Imbalanced Semi-Supervised Learning, in CVPR 2022.

---

> > ### Author Response · Authors · 2023-11-23
> > **Thank you for your response**
> >
> > Dear Reviewer hxp2,
> >
> > Thank you for your time and feedback! We provide responses to your questions below.
> >
> > ---
> >
> > ### Discussing [X. Wang et al., 2022]
> >
> > We thank the reviewer for mentioning [X. Wang et al., 2022].
> > We note that DIPS contrasts this work on several dimensions:
> > - **problem setting**: DIPS tackles the issue of label noise in pseudo-labeling. On the contrary, [X. Wang et al., 2022] assumes access to an oracle who can provide gold-truth labels for a set of initially unlabeled data. Furthermore, the terms "selective labeling" in the title of [X. Wang et al., 2022] refer to active label querying under a budget constraint, which is not the setting of DIPS (where we already have labeled and unlabeled sets defined).
> > - **modalities**: DIPS is a general purpose plugin for which we demonstrate applicability across tabular and image datasets. On the contrary, [X. Wang et al., 2022] only tackles images and requires an unsupervised representation learning step which restricts the generality of models we can use.
> >
> > We will update the Appendix to include this reference in our post-review revisions.
> >
> > ----
> >
> > ### VIME Baseline
> >
> >
> >
> > We thank the Reviewer for the suggestion, and have updated the table accordingly with the results for the supervised basline and the vanilla PL method.
> >
> > |                   | DIPS (OURS)         | VIME | Supervised | Vanilla PL |
> > |-------------------|---------------------|---------------------|----------|----------|
> > | adult             | **82.66 ± 0.10**    | 67.69 ± 0.10        |81.28±0.1 | 81.17±0.08|
> > | agaricus-lepiota  | 65.03 ± 0.25        | **66.13 ± 0.01**    |63.76±0.4 | 64.16±0.18|
> > | blog              | **80.58 ± 0.10**    | 73.52 ± 0.01        |79.27±0.08 | 79.73±0.11|
> > | credit            | **81.39 ± 0.07**    | 66.91 ± 0.02        |79.85±0.07 | 79.43±0.1|
> > | covid             | **69.97 ± 0.30**    | 68.28 ± 0.03        |67.36±0.15 | 68.26±0.14|
> > | compas            | **65.34 ± 0.25**    | 63.41 ± 0.02        |59.98±0.25 | 61.23±0.27|
> > | cutract           | **68.60 ± 0.31**    | 60.36 ± 0.04        |63.92±0.46 | 65.28±0.48|
> > | drug              | **78.16 ± 0.26**    | 74.47 ± 0.03        |74.73±0.33 | 75.34±0.37|
> > | German-credit     | **69.40 ± 0.46**    | 62.65 ± 0.05        |69.4±0.54 | 66.85±0.73|
> > | higgs             | **81.99 ± 0.07**    | 71.34 ± 0.03        |81.09±0.07 | 81.65±0.07|
> > | maggic            | **67.60 ± 0.08**    | 64.98 ± 0.01        |65.72±0.08 | 66.54±0.09|
> > | seer              | **82.74 ± 0.08**    | 80.12 ± 0.01        | 81.88±0.07 | 82.35±0.07|
> >
> > We note that VIME necessitates a neural network as it builds on pretext tasks which require training an encoder. This contrasts the framework of pseudo-labeling, which is general-purpose, and enables us to use tree-based models such as XGBoost, which are traditionally the models giving the best performance for tabular data (Grinsztajn et al., 2022), and which might explain the performance gap with the supervised baseline.
> >
> >
> > We want to highlight that DIPS aims to underscore the broader topic of labeled data quality in pseudo-labeling. Extending our experiments to include other SSL baselines could indeed provide insights, though we want to clarify that our paper's primary contribution lies in showing and addressing the overlooked aspect of labeled data quality for **pseudo-labeling**, as well as highlighting the value of DIPS' selector for this purpose, rather than conducting a comparative analysis between different SSL baselines in the tabular data setting.
> >
> >
> > Grinsztajn, Leo, Edouard Oyallon, and Gael Varoquaux. "Why do tree-based models still outperform deep learning on typical tabular data?." Thirty-sixth Conference on Neural Information Processing Systems Datasets and Benchmarks Track. 2022.
> >
> > ----
> >
> > ### Incorporation of DIPS with Non-Iteration-Based SSL
> >
> >
> >
> > We apologize if this point has not been made sufficiently clear in our previous response.
> > When integrated with FixMatch, DIPS regenerates the pseudo-labels used for the computation of learning dynamics for the unlabeled data every $K$ training steps. That means that the pseudo-labels used for the computation of the learning dynamics are frozen for each learning dynamic cycle (each one comprised of $K$ training steps). Note that this is different from the online pseudo-labels which are used to train the model: those are updated after every gradient step of the model.
> >
> > We will clarify this point in our revision.
> >
> > ---
> >
> > _Thank you for your time, please let us know if you have any other questions!_

---

> ### Comment · Reviewer_hxp2 · 2023-11-23
> **Response to authors' comment**
>
> Thank you for your feedback and further clarification. I believe there are no major issues remaining for me, and the proposed concept is both versatile and distinct from previous related literature. Therefore, I intend to maintain my favorable rating.

---

> > ### Author Response · Authors · 2023-11-23
> > **Thank you for your positive response!**
> >
> > Dear Reviewer hxp2
> >
> > Thank you! And thanks again for your time and positive feedback!
> >
> > Regards
> >
> > Paper 3725 Authors

---

### Official Review · Reviewer_yF2p · 2023-11-04

**Soundness:** 2 fair
**Presentation:** 3 good
**Contribution:** 2 fair
**Rating:** 3
**Confidence:** 4

**Summary:**

Pseudo-labeling technique is widespread semi-supervised learning nowadays. In most works it is assumed that labeled data have golden correct labels, while authors of the paper highlight that in real world cases labeled data comes with label noise. One of the main contributions of the paper is raising this issue with data centric AI perspective  and focusing on its properties and solutions (though label noise problem is known and tackled before for supervised training only). Authors propose simple yet effective selection algorithm, dubbed DIPS, which is plug and play and applicable to most pseudo-labeling algorithms: both labeled (assumed to be with label noise) and unlabeled data are selected based on both confidence and uncertainty for next teacher-student training. Uncertainty estimation is proposed to be based on the training dynamics: variation of predictions between different checkpoints. Authors validate necessity of proposed method on couple of domains: tabular data and image classification.

**Strengths:**

- Presentation of results and overall writing is of high quality
- Highlighting label noise problem in labeled data for semi-supervised learning and considering this problem from data centric AI point of view
- Proposed selection / filtering of labeled data during teacher-student process
- Results for two domains: image and tabular data with variety of datasets to show wide usage and applicability of the proposed method, as well as coverage of labeled and unlabeled data coming from different data distributions
- Robustness in the sense that different PL algorithms with proposed selection becomes close to each other in the final performance. This is very nice property as then doesn't matter what to use in practice and this speeds up development and deployment.

**Weaknesses:**

- Authors do not disambiguate the problem into two axes: i) amount of label noise ii) amount of data in labeled data. It is well known that with small amount of labeled data (not even relative to the unlabelled, but itself, say 1k-10k images, 10min-10h of speech) it is very problematic to train pseudo-labeling algorithms with good quality due to both weak initial teacher and other training dynamics. I suspect complicated dependency between label noise level and amount of labeled data (besides amount of unlabeled data) which authors do not investigate in depth.
- Absence of simple basic baselines where we apply straightforward the label noise methods to labeled data and perform standard teacher-student training. "In such situations, as shown in Fig. 1, noise propagates to the pseudo-labels, jeopardizing the accuracy of the pseudo-labeling steps" -- if we could train on small amount of labeled data with any method of learning with noisy labels then first teacher will be strong. It is not shown in the paper that all prior methods of learning with noisy labels fail on small amount of labeled data in supervised learning and thus sec 3.2 first part is overstated.
- "PL methods do not update the pseudo-labels of unlabeled samples once they are incorporated in one of the $\mathcal{D}_{train}^{(i)}$ -- authors incorrectly (check out e.g. Xie, Qizhe, et al. "Self-training with noisy student improves imagenet classification." Proceedings of the IEEE/CVF conference on computer vision and pattern recognition. 2020) formulate teacher-student pseudo-labeling widely used and thus overstated issues in second part of sec 3.2. On the next iteration of teacher-student training all unlabeled data are relabeled, some selection based on confidence or/and uncertainty is applied to pseudo-labels and then labeled data and these new pseudo-labeled data are combined (or labeled data can be skipped) to train new student model. I never saw in prior works on any teacher-student training (when new student is trained from scratch on new data) for images, text and speech input data that old pseudo-labels (from older teacher) are used along with new ones (with latest teacher).
- Absence of empirical analysis showing that confidence filtering of labeled data is not enough and aleatoric uncertainty is necessary. Also there are no baselines with widely used uncertainty-based filtering for pseudo-labeled data to be used for labeled data -- motivating usage of aleatoric uncertainty.
- Absence of empirical justification of the proposed selection method for unlabeled data only, as then we also have mislabeling and thus it should be effective there too assuming all labeled data are correctly labeled. It could be also that maybe proposed selection is only needed for labeled data while any prior selection methods could be used for unlabeled data.
- Paper conceptually messes up between teacher-student PL methods and the ones when one model continuously trains on data with time-to-time regenerated pseudo-labels by either EMA model or by previous model states. These two approaches have different training dynamics and problems (e.g. second one is less stable).
  - Training of "greedy PL" in sec. 5.1 (which is the latter type of PL methods) is out of initial formulation of PL in sec 3.1. Moreover, there is no results in Fig 3 for the zero corrupted samples, it seems PL itself it not improving upon supervised training in this toy example which is strange (looking at x=0.1)
  - Baselines in sec 5.2 are a mixture of both methods, which is not aligned again with formulation in sec 3.1.
  - Algo 1 in Appendix A does not cover the second approach, e.g. greedy PL method.

**Questions:**

- I do not agree entirely with statement "labeled data are noisy" as if labeled data are very limited in a lot of applications we could ask for the golden (correct) labels, as we need only small amount.
- "application: these works use pseudo-labeling as a tool for supervised learning, whereas DIPS extends the machinery of pseudo-labeling itself." I don't see really huge difference here, as noisy labels means - we don't know the correct label, so mathematically it is very close tasks and solutions.
- why do we use checkpoints for the learning dynamic being epoch and not some parameter based on number of iterations? For very large data we will never do 1 epoch, or only 1 epoch, or only few and thus measure based on epochs will be weak. From appendix info it is not clear even what checkpoints are selected and how this selection important.
- did authors try to exclude some first epochs from confidence and uncertainty definition as they could be not very informative?
- I don't understand this statement "Recall aleatoric uncertainty captures the inherent data uncertainty, hence is a principled way to capture issues such as mislabeling". Why does aleatoric uncertainty capture mislabeling? Why def. 4.2 is aleatoric as we consider variation across checkpoints (= over learning process)?
- What is the upper bound (when all data are labeled and when all data are labeled and correct / w/o mislabeling) in Fig. 4?
- How many iterations are done for teacher-student baselines in Fig. 4?
- This statement is incorrect "Note that s is solely used to select pseudo-labeled samples, among those which have not already been pseudo-labeled at a previous iteration." in both types of algorithms like greedy PL and UPS we relabel and reselect pseudo-labeled data.
- How the parameters of the baselines in Appendix B1 are found? Why it is not adopted per dataset? I see that aleatoric uncertainty is adopted per dataset, which could be unfair parameters selection for the baselines.
- B 3.1, 3.2 what does it mean T=5 for greedy PL and for UPS? Could authors describe exactly how they do PL for both algo with 5 iterations? Does it mean that for UPS it is 5 teacher-student trainings and for greedy PL 5 teacher-student trainings with each student training based on the original prior work where we continuously train model?
- What will happen if ablation in C2 is done only for labeled data selection and unlabeled data are selected based on PL prior works?
- What is the percentage of selected data by every method, including authors', on every iteration?
- What about Fig 11 with vanilla PL but w/o any data selection?

---

> ### Author Response · Authors · 2023-11-16
> **Response to Reviewer yF2p [Part 1/5]**
>
> Dear Reviewer yF2p
>
> Thank you for your thoughtful comments and suggestions! We give answers to each of the following points in turn and highlight the updates to the revised manuscript. In addition, we have uploaded the revised manuscript. We hope this response alleviates your concerns, but please let us know if there are any remaining concerns.
>
>
> - A) Dependency between label noise level and amount of labeled data **[Part 2/5]**
> - B) LNL + pseudo-labeling **[Part 2/5]**
> - C) Importance of aleatoric uncertainty **[Part 3/5]**
> - D) Selection of unlabeled data **[Part 3/5]**
> - E) Terminology of pseudo-labeling **[Part 4/5]**
> - F) Answers to the other questions **[Parts 4,5/5]**

---

> ### Author Response · Authors · 2023-11-16
> **Response to Reviewer yF2p [Part 2/5]**
>
> ### (A) Dependency between label noise level and amount of labeled data
> We thank the reviewer for the suggestion to investigate the dependency between label noise level and the amount of labeled data. To answer this point, we conduct a synthetic experiment following a similar setup as in Section 5.1 in our manuscript. Note that the experiment is synthetic in order to be able to control the amount of label noise.
> We considered the same list of label noise proportions, ranging from 0. to 0.45. For each label noise proportion, we consider $n_{\mathrm{lab}} \in$ {50,100,1000}, and fix $n_{\mathrm{unlab}} = 1000$.
> We conduct the experiment $40$ times for each configuration.
>
> We report the results in the plots available at: https://i.imgur.com/rzFeFsl.png.
>
> As we can see on the plots, PL+DIPS consistently outperforms the supervised baselines in almost all the configurations. When the amount of labeled data is low ($n_{lab} = 50$) and the proportion of corrupted samples is high ($0.45$), PL is on par with the supervised baseline. This mirrors the reviewer's intuition that pseudo-labeling is more difficult with a very low amount of labeled samples (and a high level of noise). We note, though, that DIPS consistently improves the PL baseline for reasonable amounts of label noise which we could expect in real-world settings (e.g. $0.1$). The performance gap between DIPS and PL is remarkably noticeable for $n_{\mathrm{lab}}=1000$, i.e. when the amount of labeled samples is high.
>
> **UPDATE:** We have included these results in Appendix C.5 of the revised manuscript.
>
> ---
>
> ### (B) LNL + pseudo-labeling
> In the following table, we report the results  when applying the LNL baselines to the labeled data only, and then performing standard teacher-student training.
>
> |                   | DIPS (OURS)           | Small-Loss | Fluctuation  | FINE |
> |-------------------|-----------------------|------------------------------|-----------------------------|----------------------|
> | adult             | **82.66 ± 0.10**      | 80.76 ± 0.20                 | 80.95 ± 0.23                | 81.06 ± 0.27         |
> | agaricus-lepiota  | **65.03 ± 0.25**      | 64.22 ± 0.27                 | 55.93 ± 1.67                | 54.61 ± 3.77         |
> | blog              | **80.58 ± 0.10**      | 79.09 ± 0.35                 | 79.16 ± 0.28                | 77.98 ± 0.72         |
> | credit            | **81.39 ± 0.07**      | 79.57 ± 0.23                 | 79.67 ± 0.26                | 77.87 ± 0.30         |
> | covid             | **69.97 ± 0.30**      | 67.76 ± 0.52                 | 67.09 ± 0.59                | 66.61 ± 0.60         |
> | compas            | **65.34 ± 0.25**      | 59.88 ± 0.46                 | 59.56 ± 0.59                | 59.94 ± 0.63         |
> | cutract           | **68.60 ± 0.31**      | 62.83 ± 1.02                 | 62.23 ± 0.93                | 63.73 ± 1.40         |
> | drug              | **78.16 ± 0.26**      | 75.19 ± 0.71                 | 74.71 ± 0.69                | 74.28 ± 1.00         |
> | German-credit     | 69.40 ± 0.46          | **71.15 ± 1.27**             | 70.95 ± 1.11                | 61.10 ± 4.95         |
> | higgs             | **81.99 ± 0.07**      | 80.91 ± 0.16                 | 80.81 ± 0.19                | 79.60 ± 0.19         |
> | maggic            | **67.60 ± 0.08**      | 65.08 ± 0.28                 | 65.18 ± 0.25                | 64.43 ± 0.33         |
> | seer              | **82.74 ± 0.08**      | 81.57 ± 0.17                 | 81.66 ± 0.25                | 79.66 ± 0.34         |
>
> The results show that DIPS  outperforms these baselines (Small-loss, Fluctuation, FINE) in almost all the datasets, and suggest that that DIPS' curation method based on learning dynamics should be preferred, both on the labeled set and on top of standard teacher-student training.

---

> ### Author Response · Authors · 2023-11-16
> **Response to Reviewer yF2p [Part 3/5]**
>
> ### (C) Importance of aleatoric uncertainty
>
> We thank the reviewer for the suggestion. We conducted an ablation study where we removed the aleatoric uncertainty and only kept a confidence-based selection (with threshold = 0.8). We term this confidence ablation to highlight if there is indeed value to the aleatoric uncertainty component of DIPS.
>
> We report results in the following table, for the $12$ tabular datasets used in Section 5.2, which shows the benefit of the two-dimensional selection criterion of DIPS. Of course, in some cases there might not be a large difference with respect to our confidence ablation --- however we see that DIPS provides a statistically significant improvement in most of the datasets. Hence, since the computation is negligible, it is reasonable to use the 2-D approach given the benefit obtained on the noisier datasets.
>
>
>
> |                   | DIPS          |  Confidence Ablation |
> |-------------------|---------------------|---------------------|
> | adult             | **82.66 ± 0.10**    | 82.13 ± 0.16        |
> | agaricus-lepiota  | **65.03 ± 0.25**        | 64.38 ± 0.23   |
> | blog              | **80.58 ± 0.10**    | 80.22 ± 0.33        |
> | credit            | **81.39 ± 0.07**    | 79.76 ± 0.15        |
> | covid             | **69.97 ± 0.30**    | 69.28 ± 0.40        |
> | compas            | **65.34 ± 0.25**    | 64.69 ± 0.25         |
> | cutract           | **68.60 ± 0.31**    | 66.32 ± 0.12         |
> | drug              | **78.16 ± 0.26**    | 75.37 ± 0.71        |
> | higgs             | **81.99 ± 0.07**    | 81.42 ± 0.16        |
> | maggic            | **67.60 ± 0.08**    | 66.26 ± 0.18          |
> | seer              | **82.74 ± 0.08**    | 82.02 ± 0.15       |
>
>
> **UPDATE:** we have included these results in Appendix C.10 of our revised manuscript.
>
>
>
> While DIPS builds on a notion of uncertainty, we highlight that this uncertainty differs in nature from the uncertainty metrics used in the pseudo-labeling literature.
> One key difference is that DIPS computes the aleatoric uncertainty based on some given labels (provided labels for $D_{lab}$ or pseudo-label for $D_{unlab}$), and then computes the agreement between the prediction of a model and these given labels as is captured by the quantity $[f_{e}(x)]_{y}$. This is very different in nature from the uncertainty metrics used in pseudo-labeling, which traditionally do not rely on a given label as they are applied to *unlabeled data* (as is done when using an ensemble of models and computing a standard deviation to define an uncertainty metric). This explains why these uncertainty based filtering methods only focus on unlabeled data, while ours can handle both labeled and pseudo-labeled data.
>
> ---
>
> ### (D) Selection of unlabeled data
>
> To answer the Reviewer's question regarding the selection of unlabeled data in DIPS, we refer to Figure 10 in our submitted manuscript (Appendix C.1). This figure shows results when our selection mechanism is only applied to the labeled data and we only use prior selection methods for the unlabeled data (this baseline is denoted as "A1"). The results show DIPS outperforms this baseline by a large margin. This justifies why we apply our selection method both to labeled and unlabeled data. We have updated our manuscript (Section 4.1) to more clearly highlight this result - thank you for the suggestion!

---

> ### Author Response · Authors · 2023-11-16
> **Response to Reviewer yF2p [Part 4/5]**
>
> ### (E) Terminology of pseudo-labeling
>
> We thank the reviewer for the comment about the terminology of teacher-student pseudo-labeling and apologize if our description of this paradigm was not clear enough in our manuscript.
> We decided to describe the common teacher-student pseudo-labeling methodology adopted in the **tabular setting**. As a consequence, we used the implementation provided by [R1], which grows the training set with pseudo-labels generated at the current iteration, thus keeping old pseudo-labels in the training dataset in subsequent iterations.
>
> In addition to adopting this practice, we investigated this choice experimentally, by comparing between two versions of confidence-based pseudo-labeling:
> - Version 1): with a growing set of pseudo-labels (as followed by the implementation of [R1] and our paper)
> - Version 2): without keeping old pseudo-labels, as suggested by the reviewer.
>
> We evaluate these two methods in the synthetic setup described in Section 5.1 of our manuscript, and report the test accuracy in the plot available at: https://i.imgur.com/VSPg1mD.png.
>
>
>
> The red line corresponds to Version 1) (the implementation we used in the manuscript), while the green line corresponds to Version 2) (suggested by the reviewer).
> As we can see, in this tabular setting, growing a training set by keeping the pseudo-labels generated at each iteration leads to the best results, motivating our adoption of this pseudo-labeling methodology used in the tabular setting.
>
> **UPDATE:** we have included this experiment in Appendix C.6 of our revised manuscript.
>
>
> We apologize if the terminology of greedy-PL was not made sufficiently clear in our original manuscript. We refer to confidence-based PL as "greedy-PL", following the same terminology as in [R1]. As such, all the baselines in Section 5.2 fall under the umbrella of " teacher-student PL methods", where the models (i.e. XGBoost in Section 5.2) are trained from scratch at each iteration in a teacher/student fashion.
>
>
> Finally, to address the query about Fig 3 with zero corrupted samples, we add to the results in Fig.3 the zero corruption setting in the updated plot available at: https://i.imgur.com/EBUYDkY.png.
>
> In the zero corruption setting, all methods almost achieve perfect test performance.
>
> **UPDATE:** we have updated Fig. 3)c) in our revised manuscript.
>
>
> We also highlight that pseudo-labeling does always increase performance over the purely supervised approach, which is natural since PL has access to unlabeled data. However, we are interested in a setting where the labeled data is subject to **mislabeling**. Hence, this explains why the margin between PL and Supervised in Fig. 3)c) is not bigger: noise propagates to the unlabeled data, which limits the effectiveness of PL and justifies the data-centric lens that DIPS adopts, hence giving a big performance gap with the baselines.
>
> ---
>
> ### (F) Answers to the other questions
> - *"I do not agree entirely with statement "labeled data are noisy \& if small amounts we can request the gold standard"*
>
> The issue of labeled data errors are prevalent in many industries from healthcare to finance. This is even the case in seemingly curated datasets as shown by Northcutt et al., where label error rates of widely-used benchmark datasets can reach up to 10\%. As mentioned in Section 1, while it might appear possible to manually inspect the data to identify errors in the labeled set, we note that this requires domain expertise and is human-intensive, especially in modalities such as tabular data where inspecting rows in a spreadsheet can be much more challenging than reviewing an image. In other cases, updating labels is actually infeasible due to rerunning costly experiments in domains such as biology and physics, or indeed impossible due to lack of access to either the underlying sample or equipment.
>
> - *"application: these works use pseudo-labeling as a tool for supervised learning, whereas DIPS extends the machinery of pseudo-labeling itself." I don't see really huge difference here, as noisy labels means - we don't know the correct label, so mathematically it is very close tasks and solutions.*
>
> We agree that mathematically, the auditing task is similar. The main difference is how semi-supervised learning is used. In the first setting SSL is a solution to noisy labels, whereas we study and resolve the effect of noisy labels on SSL.

---

> ### Author Response · Authors · 2023-11-16
> **Response to Reviewer yF2p [Part 5/5]**
>
> - *"why do we use checkpoints for the learning dynamic being epoch and not some parameter based on number of iterations? For very large data we will never do 1 epoch, or only 1 epoch, or only few and thus measure based on epochs will be weak."*
>
> We first note that DIPS only needs models which are trained in an iterative manner. For example, in our tabular experiments in Section 5.2, we use an XGBoost backbone, where the learning dynamics are computed over the different boosting iterations. Hence what we call a checkpoint in our manuscript (Section 4.2) needs not be an epoch. For neural networks, we do not necessarily need to define checkpoints as epochs, when the dataset is large; we can instead define checkpoints after every $k$ gradient steps.
>
> - *"did authors try to exclude some first epochs from confidence and uncertainty definition as they could be not very informative?"*
>
> We thank the reviewer for this suggestion. We have conducted an additional experiment to investigate the choice of the range of iterations used to compute the learning dynamics. We consider ignoring the first $25$%/$50$%/$75$% iterations and use the remaining iterations to compute the learning dynamics.
>
> The plot available at https://i.imgur.com/QzNnHba.png
>
> It shows the mean performance difference (on test accuracy) by using the truncated iteration windows versus using all the iterations, and averages the results over the 12 datasets used in Section 5.2. As we can see, it is better to use all the iterations window, as the initial iterations carry some informative signal about the hardness of samples.
> This motivates our choice of computing the learning dynamics over the whole optimization trajectory, a choice which we adopt for all of our experiments.
>
> **UPDATE:** we have added these experimental results in Appendix C.7 of our revised manuscript.
>
>
> - *"Why does aleatoric uncertainty capture mislabeling? Why def. 4.2 is aleatoric as we consider variation across checkpoints (= over learning process)?"*
>
> Def. 4.2 of aleatoric uncertainty builds on the variability that originates from the inability to predict the correct label with high confidence over the different checkpoints. Intuitively, a sample with high aleatoric uncertainty would be a sample for which we are not confident in predicting the ground-truth label at every checkpoint (e.g. $[f_{e}(x)]_{y} = 1/2$ for all iteration $e$ in the case of binary classification). This would mean that the model is not capable of learning the given label, which happens for mislabeled points.
>
> - *"What is the upper bound (when all data are labeled and when all data are labeled and correct / w/o mislabeling) in Fig. 4?"*
>
> We thank the reviewer for the question and we report in the following plot, available at https://i.imgur.com/gsLrb7r.png, the test accuracy when all the data are labeled in Section 5.2. Note that we cannot control the amount of mislabeling in this setting, as we only have access to the training labels in the datasets.
>
>
> Training a model on all the labeled data often yields the best results, which is intuitive, as we consider the pseudo-labeling setting which uses only $10\%$ of labeled data. We notice that in some datasets (e.g. cutract, compas, drug), DIPS improves the fully supervised baseline, which may hint at greater inherent noise in the datasets. We can't quantify exactly the level of noise as we don't have the ground-truth, but the performance gap obtained by using DIPS could be seen as a proxy for it.
>
> - *"How many iterations are done for teacher-student baselines in Fig. 4?"*
> *"I see that aleatoric uncertainty is adopted per dataset."*
>
> We used $T=5$ iterations for the experiment in Fig. 4. This follows the guideline of the sensitivity experiment conducted in [R1] (Appendix B.2 in the corresponding paper) and explains why we kept this parameter fixed for the datasets used in Section 5.2 in our manuscript.
> Regarding the aleatoric uncertainty threshold, it is chosen to be adaptive, and is not hand-crafted for each dataset. Furthermore, we stress that DIPS is applied on top of the baselines.
>
>
>
> - *"What about Fig 11 with vanilla PL but w/o any data selection?*"
>
> We give the radial diagram showing the target dataset and how the DIPS selector and vanilla PL selector differ at the following link: https://i.imgur.com/f4VV76N.png.
>
> We see that DIPS is closer to the test data than vanilla PL selector.
>
> **UPDATE:** we have added this plot in Appendix C.3 in our revised manuscript.
>
> We hope this answers your points, please let us know if there are any remaining concerns.

---

> ### Author Response · Authors · 2023-11-16
> **References**
>
> ### References
> [R1] Vu-Linh Nguyen, Sachin Sudhakar Farfade, and Anton van den Hengel. Confident Sinkhorn allocation for pseudo-labeling. arXiv preprint arXiv:2206.05880, 2022.

---

> ### Author Response · Authors · 2023-11-20
> **Dear Reviewer yF2p**
>
> Dear Reviewer yF2p
>
> We are sincerely grateful for your time and energy in the review process.We hope that our responses and appendix/manuscript updates have been helpful. Please let us know of any leftover concerns and if there was anything else we could do to address any further questions or comments!
>
> Thank you!
> Paper 3725 Authors

---

> > ### Comment · Reviewer_yF2p · 2023-11-21
> > **Response to the authors' comments and updated manuscript**
> >
> > Dear authors,
> >
> > I have read carefully all reviews and your responses as well as updated manuscript. First, thanks a lot for all detailed responses and extra ablations. Second, please find below my further comments and questions:
> >
> > > comparison to VIME
> >
> > I have looked at this ablation too in the revised manuscript. Results with VIME are worse than supervised baseline (e.g. seer, adult, credit and others) in Figure 4, why is that? Seems VIME experiments could be incorrect then. Please correct me if I missed anything here.
> >
> > >  (A) Dependency between label noise level and amount of labeled data
> >
> > Thanks for conducting these experiments. For future, I think overall running 10 times is enough for getting good estimate of the std from experiments + running on real, not synthetic data, will be more helpful from the application side (e.g. you can control in CIFAR-100 how many labels you flip randomly). But ok, I am fine with the setting for the synthetic data.
> >
> > From the results in Figure 17 now I could  infer that 1) for small amount of labeled data it is less helpful (as I expected it is just very hard setting); 2) I can simply filter out noisy samples from labeled data (using any prior  method on identifying noisy samples in data), use 2x less data (1000 -> 500) and then I get the best model with classic PL / no PL at all (as here with clean labeled data we even don’t need to use unlabeled data).
> >
> > Main question from the results I have right now is "what if I filter out noise from labeled data first and then train standard supervised baseline and PL baseline? would they be better than DIPS?
> >
> > > (B) LNL + pseudo-labeling
> >
> > Thanks for pointing to the previously done ablation. Sorry that I missed it in the initial review. I think I misunderstood that section in the Appendix, now I got what you meant. Could you also provide supervised only (w/o doing PL) for all the methods from the Table here as well as the baseline trained with standard cross-entropy loss only, as I see that results for these prior methods are worse than supervised baseline from Figure 4 (e.g. adult and seer). The latter is very suspicious that PL is hurting the results (especially in the context that supervised baselines with, say FINE, should be even better than one from Figure 4)?
> >
> > > (C) Importance of aleatoric uncertainty
> >
> > Thanks for the ablation! Having these results I believe Figure 4 should be revisited in the paper, and the Vanilla runs should be done with confidence filtering included: it was shown in many prior works already that confidence filtering is needed and thus it is more fair comparison. You propose to add aleatoric uncertainty and thus exactly improvement on top of confidence filtering should be shown in Figure 4. Right now it is very misleading for results and overstating the improvements the paper brings. I believe if you redo Figure 4 with that you will have a less “impressive” results as highlighted by Reviewer `R-hxp2`.
> >
> > > while ours can handle both labeled and pseudo-labeled data.
> >
> > I would argue here that uncertainty metrics focused on unlabeled data still can be applied to labeled data as they do not need any labels and we can ignore labels in labeled data.
> >
> > > (D) Selection of unlabeled data
> >
> > Either I still don’t understand formulation or something is incorrectly formulated in the text. A1 is when you do selection only for training on labeled data. A2 is doing selection for both labeled and unlabeled data during next phases of training. There is no baseline when you do selection **only on labeled data** and do not do anything with unlabeled data (or do confidence filtering as in prior works). Moreover, what I proposed initially is to do experiments where you don’t have any label noise in labeled data and then apply DIPS on unlabeled data only, to show that previously proposed confidence filtering is not enough for unlabeled data (even in the setting of no label noise in labeled data). This I believe should work as we can interpret pseudo-labels as data with noise.

---

> > > ### Comment · Reviewer_yF2p · 2023-11-21
> > > **Response to the authors' comments and updated manuscript [continue]**
> > >
> > > > (E) Terminology of pseudo-labeling
> > >
> > > Provided reference [R1] is not published in the peer-reviewed journals / conferences, thus I would suggest to use prior published papers definitions as a reference or introduce new variant of pseudo-labeling and then make connection to prior works / do proper comparisons / justify why we need another definition-algorithm.
> > >
> > > Compared baselines on FixMatch, UPS, FlexMatch, etc. do not use such definition on growing set of pseudo-labels in their experiments, moreover even 1) do continuous training with selecting new batch of unlabeled data each iteration and generating new PLs or 2) do teacher-student training. It is not clear then what authors mean exactly when they point to a particular paper baseline implementation and what is used. If they use their growing pseudo-label set — it is another algorithm than in prior works. Right now I don’t have any understanding how the baseline methods are implemented. I believe in every baseline, proposed filtering based on confidence + uncertainty should be used, not another modification like expanding set of pseudo-labels (which includes older generated versions).
> > >
> > > > The red line corresponds to Version 1) (the implementation we used in the manuscript), while the green line corresponds to Version 2) (suggested by the reviewer). As we can see, in this tabular setting, growing a training set by keeping the pseudo-labels generated at each iteration leads to the best results, motivating our adoption of this pseudo-labeling methodology used in the tabular setting.
> > >
> > > I appreciate! However, this should be tested for every baseline/dataset, not only for synthetic data: it will be very dependent on the method of PL you use. Right now I see that you introduce another algorithms on PL which is non-standard in the literature (thus could be a new contribution) and is not tested e.g. on images. It maybe used in [R1] but there is no justification why it is better than prior works. Experiments on synthetic data only with on PL variant is not enough in y opinion (this even can be a separate paper).
> > >
> > > >We apologize if the terminology of greedy-PL was not made sufficiently clear in our original manuscript. We refer to confidence-based PL as "greedy-PL", following the same terminology as in [R1]. As such, all the baselines in Section 5.2 fall under the umbrella of " teacher-student PL methods", where the models (i.e. XGBoost in Section 5.2) are trained from scratch at each iteration in a teacher/student fashion.
> > >
> > > Thanks for clarification. So this does not follow your definition in Section 3. All variants which are used, training from scratch or continue model training (this one can be also used in xgboost, where every new tree in the ensemble is treated as training step like in NNs), should be clearly specified in the paper because, as I said, they have different properties and dynamics in practice.
> > >
> > > greedy-PL in the text via Lee et al., 2013 is continuous training, which is different from your reference to [R1]. I would suggest be clear in the text that for tabular data (as I understand) you use only teacher-student training and you repeat 5 times the process with increasing number of pseudo-labels for every unlabeled sample. For images, probably you do different training as FixMatch is not the training of teacher-student type, it is continuous training.
> > >
> > > > Finally, to address the query about Fig 3 with zero corrupted samples, we add to the results in Fig.3 the zero corruption setting in the updated plot available at: https://i.imgur.com/EBUYDkY.png. In the zero corruption setting, all methods almost achieve perfect test performance.
> > >
> > > I would argue that this setting seems to be unrealistic and the question could be “does the behavior of all ablations on such synthetic data can be extrapolated to the cases where zero corruption case actually have huge gap between supervised and semi-supervised training?” The effect of having labeled data corruption can be even bigger in practice as we rely a lot on unlabeled data then.. Frankly, I think this can be more strong argument in your favor if we have data where for zero corruption there is difference between supervised an semi-supervised training.
> > >
> > > > Baselines in sec 5.2 are a mixture of both methods, which is not aligned again with formulation in sec 3.1. Algo 1 in Appendix A does not cover the second approach, e.g. greedy PL method.
> > >
> > > This is still an issue for me.
> > >
> > > > We agree that mathematically, the auditing task is similar. The main difference is how semi-supervised learning is used. In the first setting SSL is a solution to noisy labels, whereas we study and resolve the effect of noisy labels on SSL.
> > >
> > > I would suggest to smooth a bit formulation.

---

> > > > ### Comment · Reviewer_yF2p · 2023-11-21
> > > > **Response to the authors' comments and updated manuscript [continue]**
> > > >
> > > > > We first note that DIPS only needs models which are trained in an iterative manner. For example, in our tabular experiments in Section 5.2, we use an XGBoost backbone, where the learning dynamics are computed over the different boosting iterations. Hence what we call a checkpoint in our manuscript (Section 4.2) needs not be an epoch. For neural networks, we do not necessarily need to define checkpoints as epochs, when the dataset is large; we can instead define checkpoints after every gradient steps.
> > > >
> > > > I see main confusion in the way you define iterative. Is it continuous training of the model (like, grad steps in NNs, adding one more tree into ensemble in the xgboost) or it is resetting model and training from scratch? I feel you mix this terminology between xgboost and NNs, and between prior works (as in most of them training is done continuously, like FixMatch). This creates a lot of interpretations what and how exactly you are doing in the paper and experiments right now. Not clear how overall proposed data selection influence both variants of PL and how the choice of the checkpoints affects the dynamics and thus the selection criteria.
> > > >
> > > > > We thank the reviewer for this suggestion. We have conducted an additional experiment to investigate the choice of the range of iterations used to compute the learning dynamics. [...]
> > > >
> > > > Thanks for the ablation. So if I correctly interpret PL algorithm here — it is xgboost where each iteration is teacher-student training. Then yes, I agree it makes sense to use all, as even first iteration here is well trained already and properly showing the uncertainty of the model/data (Figure 2 then is really misleading for the xgboost then). However if we come back to FixMatch training on CIFAR where iteration will be another grad step — then I guess initial iterations will be misleading.
> > > >
> > > > > This would mean that the model is not capable of learning the given label, which happens for mislabeled points.
> > > >
> > > > What then happens if for samples we have memorization in the model?
> > > >
> > > > > We thank the reviewer for the question and we report in the following plot, available at https://i.imgur.com/gsLrb7r.png, the test accuracy when all the data are labeled in Section 5.2. Note that we cannot control the amount of mislabeling in this setting, as we only have access to the training labels in the datasets.
> > > >
> > > > Thanks for adding this! This is helpful! I think the plot in the paper is not updated (but it ok, I checked the link which shows results correctly).
> > > >
> > > > > We used iterations $T=5$ for the experiment in Fig. 4
> > > >
> > > > Thanks for clarification! But then what about CIFAR / images experiments?
> > > >
> > > > > We give the radial diagram showing the target dataset and how the DIPS selector and vanilla PL selector differ at the following link: https://i.imgur.com/f4VV76N.png.
> > > >
> > > > Thanks!

---

> ### Author Response · Authors · 2023-11-23
> **Thank you for your response [Part 1/2]**
>
> Dear Reviewer yF2p,
>
> Thank you for your time and questions! We answer your questions in what follows.
>
> **On VIME results**: In order to get these results, we used the code repository released by the authors of VIME (https://github.com/jsyoon0823/VIME) to avoid any difference in implementation. We note that VIME necessitates a neural network as it builds on pretext tasks which require training an encoder. This contrasts the framework of pseudo-labeling, which is general-purpose, and enables us to use tree-based models such as XGBoost, which are traditionally the models giving the best performance for tabular data (Grinsztajn et al., 2022), and which might explain the performance gap with the supervised baseline.
>
> Grinsztajn, Leo, Edouard Oyallon, and Gael Varoquaux. "Why do tree-based models still outperform deep learning on typical tabular data?." Thirty-sixth Conference on Neural Information Processing Systems Datasets and Benchmarks Track. 2022.
>
>
> ----
>
> **Filtering out noise from labeled data + training PL baseline:** We apologize if this was not clear: as mentioned in our previous response, filtering out noise from labeled data first and training the PL baseline (which is *confidence-based filtering*, and does not use DIPS) corresponds to the A1 baseline in Appendix C.1. As we show in Figure 10, DIPS outperforms A1.
>
> ----
>
> **On LNL results**: The Reviewer is right that some of the results in Appendix C.2 are worse than the supervised baseline. We attribute this to the unsuitability of these selectors for the setting DIPS tackles. While these methods are part of the LNL literature, they traditionally assume access to large labeled datasets, which contrast our pseudo-labeling setting, in which none of these baselines have been used before. Hence, there are no guarantees of performance translations during the iterative procedure.
>
> ---
>
>
> **On adding confidence filtering:** We want to emphasize that the baselines we consider in Figure 4 _already_ use confidence filtering in their selection of pseudo-labels. For example, greedy-PL solely uses a confidence threshold, while UPS adds to it an uncertainty threshold. We integrate DIPS on top of them in the experiment of Figure 4.
>
>
> ---
>
>
>
> **Uncertainty metrics designed for unlabeled data applied to labeled data:** We wish to clarify that the uncertainty metrics focused on unlabeled data necessitate us to train a supervised model in the first place. Thereafter, we predict on the unlabeled set and apply the uncertainty metric to decide which pseudo-labels to keep. This is unsuitable for the labeled data as we would then have to use the same supervised model to predict on the very same labeled data --- yet having to "ignore" the label of the labeled sample.
> For example, in tabular data, the uncertainty proxy is often obtained with an ensemble. However, training this ensemble on labeled data and evaluating it on the same labeled data would reduce diversity in the predictions of the ensemble and lead to bad calibration of this uncertainty on labeled data (with the ensemble being very overconfident).
>
>
> ---
>
>
> **On ablation in Appendix C.6:** We will conduct these experiments on the datasets of Section 5.2 for our post-review revisions considering the limited time before the end of the rebuttal window.
>
> ---

---

> ### Author Response · Authors · 2023-11-23
> **Thank you for your response [Part 2/2]**
>
> **Baseline implementations:** The implementation of the baselines in our tabular experiments (Section 5.2) trains the models from scratch, while FixMatch in our images experiment (Section 5.5) is continuous model training. Hence, the methods in Section 5.2 are not implemented as a mixture of these two types of methods, they are implemented as teacher-student methods. We used their implementation provided by the authors of [R1] at the following link: https://github.com/amzn/confident-sinkhorn-allocation. We will clarify this distinction between teacher-student training and continuous model training in our revised manuscript.
>
> ---
>
>
> **On iterations for learning dynamics**: Let us clarify: in Section 4.2, we define the learning dynamics by studying how the model predicts samples over different **checkpoints**. For example, for a neural network, a checkpoint can be an epoch. For an XGBoost, this can be a boosting iteration.
>
> These checkpoints are not to be confused with the teacher-student iterations of the baselines that we use in Figure 4 and Section 5.2.
>
>
>
> ---
>
> **On ablation in Appendix C.7:** In the ablation of Appendix C.7, the "iterations" denote the boosting iterations of the XGBoost model. For example, 25% (on the x-axis of the plot) means that in order to compute the learning dynamics of the XGBoost model trained at _every_ teacher-student iteration, we use the last 75% of the boosting iterations of the XGBoost.
>
> ----
>
>
>
> **On memorization in the model:** Memorization may impact the computation of the learning dynamics. However, with learning dynamics we compute the metrics by looking at the behavior of the model over multiple checkpoints, and not only the last one. In the latter case, memorization would indeed be a problem. Furthermore, we want to point out to theoretical results that show (in some simple scenarios) that mislabeled points are unlikely to be classified with their (incorrect) label. For example, Theorem 3 in (Maini et al., 2022) (Appendix A.4 in the corresponding paper) gives some intuition about mislabeled samples in a mixture setting, by stating that the probability that mislabeled examples are classified with their given (incorrect) labels tends to $0$, as the model is trained until convergence.
>
>
> ---
>
> **FixMatch+DIPS parameters:** For the image experiments, we define one checkpoint every $1024$ gradient steps, and compute the learning dynamics over $10$ checkpoints.
>
> ---
> We thank you for your time and energy in the reviewing process, please let us know if you additional questions!
>
> ---
> **References:**
>
> Maini, Pratyush, Saurabh Garg, Zachary Lipton, and J. Zico Kolter. "Characterizing datapoints via second-split forgetting." Advances in Neural Information Processing Systems 35 (2022): 30044-30057.

---

> > ### Comment · Reviewer_yF2p · 2023-11-23
> > **Further discussion**
> >
> > Dear authors,
> >
> > Thanks for additional clarifications! I need to re-read the paper again and think more about results, based on your comments and our fruitful discussion.
> >
> > One thing, for sure, will be helpful for me for the final decision if you could clarify the exact setup for every baseline you run and corresponding modification when you run DIPS on top of it. Right now I am really lost in the formulations you use for pseudo-labeling. E.g. you mentioned that FixMatch for tabular data is very different from the FixMatch for images (I would argue here it is not FixMatch anymore, honestly, for tabular data then). I totally understand that particular methods maybe were designed or tested for a specific domain and are not right away applicable e.g. to tabular data and switch from NNs to XGboost. But I want you to be very specific in formulation how exactly models/baselines are trained and modified from the original papers to be sure 1) I fully understand it and then can judge on the validity of the comparisons and experiments you did; 2) we could converge to the version for the revision which will be readable for any person out of pseudo-labeling domain to use your method in practice; 3) for the sake of reproducibility.
> >
> > Thanks in advance!

---

> ### Author Response · Authors · 2023-11-23
> **Thank you for your response**
>
> Dear Reviewer yF2p,
>
> Thank you for your engagement during this discussion period!
> Let us clarify in this response the baselines used in our experiments.
>
> ## In Sections 5.1, 5.2, 5.3, 5.4 (Tabular experiments)
>
> | Name of PL method                                   | Type of PL implementation         | Backbone model | Pseudo-label selection method                              | Number of teacher-student iterations |
> | --------------------------------------------------- | ------------------------- | ---------------| ----------------------------------------------------- | ------------------------------------- |
> | Greedy-PL (also abbreviated "PL" in Figure 4)       | Teacher-student method    | XGBoost         | Confidence thresholding                              | 5 iterations                         |
> | UPS                                                 | Teacher student-model     | XGBoost         | Confidence thresholding + Uncertainty thresholding (with ensembles) | 5 iterations                         |
> | FlexMatch                                           | Teacher-student method    | XGBoost | Adaptive confidence thresholding (with Curriculum learning) | 5 iterations                         |
> | SLA                                                 | Teacher-student method    | XGBoost         | Confidence-based assignment with Optimal transport         | 5 iterations                         |
> | CSA                                                 | Teacher-student method    | XGBoost           | Confidence-based assignment and uncertainty (with ensembles) | 5 iterations                                   |
>
> For all these baselines: we use the implementation provided by https://github.com/amzn/confident-sinkhorn-allocation.
> Also note that all these methods are implemented as **teacher-student models** in the tabular experiments of Sections 5.1, 5.2, 5.3, 5.4, and that we do _not_ use FixMatch for these tabular experiments.
>
> **DIPS integration**: DIPS is integrated as a plug-in on top of these baselines, for **both labeled samples and pseudo-labeled samples selected by these baselines, _before each_ teacher-student iteration.**
> Since DIPS is a plug-in, it means that these baselines still select pseudo-labeled samples based on their respective unlabeled data selection mechanism (e.g. "Confidence thresholding"). DIPS is simply applied on top of them, for both labeled and pseudo-labeled data. That is why we refer to PL+DIPS for example in Figure 5 (i.e. we use the first baseline in the above table and plug-in DIPS on top). Similarly, in Figure 4, on the grouped bar plots, DIPS is plugged on top of each baseline (giving PL+DIPS, UPS+DIPS, etc.).
>
> ---
> ## In Section 5.2 (Image experiment)
> |Name of PL method | Type of PL implementation | Backbone model |  Pseudo-label selection method | Number of training steps |
> |------|------|-----|-----|------|
> |FixMatch| Continuous-training method |  WideResnet-18| Confidence thresholding| $1024 \times 10^3$ gradient steps
>
>
> **DIPS integration**: DIPS updates the learning dynamics at every checkpoint. We define one checkpoint every $1024$ gradient steps. DIPS performs its selection of labeled and unlabeled data every $10$ checkpoints. For the unlabeled data, DIPS uses the pseudo-labels computed by the model at the beginning of every learning dynamics cycle. Then, after every $10$
> checkpoints, DIPS curates the labeled and unlabeled data using its selector function and then regenerates the pseudo-labels used for the computation of learning dynamics on unlabeled data for the next learning dynamics cycle.
>
> Note that the frozen pseudo-labels used for the learning dynamics are not to be confused with the online pseudo-labels which are used to train the WideResnet model: those are updated after every gradient step of the model.
>
>
> _We hope this clarifies your point, please let us know if you have any other questions!_

---

> > ### Comment · Reviewer_yF2p · 2023-11-23
> > **Further discussion**
> >
> > Dear authors,
> >
> > Thanks for details! One last clarification I need before I re-read the paper to make a final decision.
> >
> > In all examples you described for teacher student you reuse previously generated pseudo-labels, so that you could have in the training examples repeated samples but with different pseudo-labels generated by different teacher-student iteration. Correct?
> >
> > For the FixMatch on images in Sec 5.2 you do not do this as it is continuous and we always regenerate PL with the current model state for the batch we train on right now. Correct?
> >
> > Thanks again for all clarifications!

---

> > > ### Author Response · Authors · 2023-11-23
> > > **Thank you for your response**
> > >
> > > Dear Reviewer yF2p,
> > >
> > > Thank you for your engagement and prompt response! We really appreciate it.
> > >
> > >
> > > * On teacher-student:
> > > For the teacher-student baselines, we do not repeat the training examples and do not assign _different_ pseudo-labels to them. After the selection, a pseudo-label is assigned to samples from the unlabeled set and added to the labeled set. Once the samples are in the labeled set, we do not assign them alternative labels, as they are discarded from the unlabeled set.
> > >
> > >
> > > * On FixMatch:
> > > You are right, since FixMatch is a continuous-training method, we do not have this cache mechanism, as the model updates the PL at every batch before doing the gradient step. Note though that to compute learning dynamics we need to freeze pseudo-labels at the beginning of each learning dynamics cycle.
> > >
> > > _We hope these clarifications are useful! Thanks again for all the engagement._

---

> > > > ### Comment · Reviewer_yF2p · 2023-12-01
> > > > **Final version of the paper feedback**
> > > >
> > > > Dear authors,
> > > >
> > > > I do appreciate all discussion and hard efforts, as well as all empirical results and analysis. I had another deep reading and digging into all our (including other reviewers) discussion and latest paper version.
> > > > - First, the paper needs revision on the notation, definition and references:
> > > >   - Equation (1) is incorrect as on every iteration you consider $\mathcal{D}\_{unlab} \setminus \mathcal{D}^{i}$ and not $\mathcal{D}\_{unlab}$. This confusion creates issues on clear reading and understanding the process you are doing, e.g. growing set of pseudo-labeled data with pseudo-labels generated by previous steps and not updated with later iterations. Next, you need to highlight that actually you are doing initial selection on the labeled data as $\mathcal{D}^1 = \mathcal{D}^1\_{train}$, so part of labeled data which you filter here will be never used during training again. (this could be all noise filtering).
> > > >   - "Second, PL methods do not update the pseudo-labels of unlabeled samples once they are incorporated in one of the $\mathcal{D}\_{train}^{(i)}$" -- this is factual error, as for FixMatch in vision (when method is actually continuous pseudo-labeling) this does not hold.
> > > >   - "Greedy pseudo-labeling (PL) (Lee et al., 2013)" --  used incorrectly, as Lee et al. a) is not teacher-student (it is continuous training); b) did not use confidence filtering; c) confidence of two bounds ($\tau_n$ and $\tau_p$) was not used there too (even I believe definition on page 15 is wrong as lower/upper confidence is applied for the same class).
> > > >   - based on Eq 1 it is clear that DIPs is applied on top of the PL selector, however it is better to highlight this in Fig 4 by marking all DIPS as "+DIPS".
> > > >   - FixMatch is not articulated at all in the paper. Explanation of how exactly it is done is still not clear and ambiguous (the more times I read your explanation, the more I don't understand how it is done).
> > > > - Second, I believe there is incorrect usage of all prior methods as the baselines, on top of which DIPS is applied. "Note that s is solely used to select pseudo-labeled samples, among those which have not already been pseudo-labeled at a previous iteration." This is not the way it is done in prior work, like Lee, FixMatch, UPS, FlexMatch, SLA (at least what I understood from the paper). The whole set of unlabeled data is considered at every iteration and a model is used to relabel all data and select the most confident. Right now it artificially creates conditions for DIPS to work, as data at each iteration contains samples with old pseudo-labels (from older iterations) which of course we need to filter (what DIPS is doing). You are referring to "Vu-Linh Nguyen, et al." -- but this the only work which is not peer-reviewed and was doing growing set of pseudo-labeled data. I think this is a technical issue when you modify prior works while following only one work, + having old pseudo-labels which of course are not suited over time. This growing set of pseudo-labels does not make sense to me. Thus I think the serious flaw of the current state of the paper is the lack of proper PL baselines and formulations.
> > > > - You followed up on this in the discussion, and did ablation in C6. When I re-read it, what I interpret based on your comments is that "Version 2): without keeping old pseudo-labels." corresponds to the case where you excluded samples with older pseudo-labels, but new data are obtained from the remaining part of unlabeled data only. This does not correspond to the standard algorithm used in prior works. Moreover it somewhat confirms intuition, that DIPS probably filters out all old pseudo-labeled data and keeps only newer ones.
> > > > - Performance issue pointed out by other reviewers is still there, as we reiterate all checkpoints on all data. Even if convergence is faster it is not clear how this scales to huge amounts of unlabeled data -- realistic cases we have nowadays.
> > > > - "On LNL results: The Reviewer is right that some of the results in Appendix C.2 are worse than the supervised baseline. We attribute this to the unsuitability of these selectors for the setting DIPS tackles. While these methods are part of the LNL literature, they traditionally assume access to large labeled datasets, which contrast our pseudo-labeling setting, in which none of these baselines have been used before. Hence, there are no guarantees of performance translations during the iterative procedure." It is really not articulated at all in the paper that the goal is small scale data (at least for unlabeled data). This part of empirical results is still very unclear to me + paper needs another positioning then.
> > > >
> > > > I believe I could miss the points and explanations, but this also means that better paper presentation and explanation are needed in the text. Based on the above summary, I will keep my score unchanged and encourage authors to revisit notation, formulations and standard baselines in PL to strengthen the paper in future.

---

### Author Response · Authors · 2023-11-16
**Response Overview**

We thank the Reviewers for their insightful and positive feedback, and their time during the review process!


We are encouraged the Reviewers found that our identified research gap highlighting the impact of label noise in labeled data for semi-supervised learning was an “interesting problem” (``R-K9EF``),  “well-justified research gap, addressing the often overlooked errors in labeled data” (``R-hGhE``) and provides a “useful data-centric AI point of view” (``R-yF2p``). Further, we are glad the reviewers deemed DIPS a “simple and effective solution” (``R-K9EF``), which is “impactful” (``R-hGhE``) and the effectiveness “impressive” (``R-hxp2``). Our method's broad applicability and consistent performance improvements across diverse datasets, including real-world image and tabular data, were particularly highlighted (``R-hxp2, R-yF2p, R-hGhE, R-K9EF``). Further, DIPS could speed up development and deployment in practice (``R-yF2p``) and have a “positive impact on the practical application of SSL in the real world” (``R-K9EF``).

We address specific questions and comments below.

We have updated our manuscript with:

- additional ablations
- new experiments (e.g. additional image datasets and VIME baseline).

We include these in Appendix C. Furthermore, all updates in the revised manuscript are colored in purple.


On the basis of our clarifications and updates, we hope we have addressed the Reviewers' concerns. Thank you for your kind consideration!

Paper 3725 Authors

---

### Comment · Area_Chair_KLLo · 2023-11-17
**Author-Reviewer Discussion Phase**

Thank you, reviewers, for your work in evaluating this submission. The reviewer-author discussion phase takes place from Nov 10-22.

If you have any remaining questions or comments regarding the rebuttal or the responses, please express them now. At the very least, please acknowledge that you have read the authors' response to your review.

Thank you, everyone, for contributing to a fruitful, constructive, and respectful review process.

AC

---

### Meta-Review · Area_Chair_KLLo · 2023-12-04

**Metareview:**

This paper considers the problem where labeled data may not be correctly labeled in semi-supervised learning. The authors propose a new method called DIPS, which leverages training dynamics to select reliable labeled data and pseudo-labels for unlabeled data. The authors validate their proposal through experiments on tabular and image data. However, there are concerns regarding the overall presentation and experimental validity. The authors aim to propose a framework that covers both tree models and neural network models for tabular and image data. However, the current presentation may confuse readers due to the existing differences between these models. This aspect needs to be revised carefully. The authors should provide a comprehensive review of previous pseudo-labeling-based methods as a foundation for their proposal. Additionally, concerns are raised about the correct implementation of experiments, particularly for the neural network-based baseline methods. Although the authors have made progress in providing a robust SSL method for imperfect labeled data, it is recommended that they revise the manuscript to eliminate misunderstandings and improve the work based on a thorough review of previous pseudo-labeling-based SSL methods. Therefore, I recommend rejecting this paper.

**Justification For Why Not Higher Score:**

The current presentation is likely to confuse readers, and to some extent, it differs from existing pseudo-labeled SSL terminology. The authors should strengthen the review and discussion of literature. The description of the experimental methods in the current study is likely to confuse readers. The authors should further clarify the specific implementation of the experiments, especially regarding different modal data and different backbone models.

**Justification For Why Not Lower Score:**

N/A

---

### Decision · Program_Chairs · 2024-01-16

Reject